# CERTIFIED ROBUSTNESS FOR FREE IN DIFFERENTIALLY PRIVATE FEDERATED LEARNING

## ABSTRACT

Federated learning (FL) provides an efficient training paradigm to jointly train a global model leveraging data from distributed users. As the local training data comes from different users who may not be trustworthy, several studies have shown that FL is vulnerable to poisoning attacks where adversaries add malicious data during training. On the other hand, to protect the privacy of users, FL is usually trained in a differentially private manner (DPFL). Given these properties of FL, in this paper, we aim to ask: *Can we leverage the innate privacy property of DPFL to provide robustness certification against poisoning attacks? Can we further improve the privacy of FL to improve such certification?* To this end, we first investigate both user-level and instance-level privacy for FL, and propose novel randomization mechanisms and analysis to achieve improved differential privacy. We then provide two robustness certification criteria: *certified prediction* and *certified attack cost* for DPFL on both levels. Theoretically, given different privacy properties of DPFL, we prove their certified robustness under a bounded number of adversarial users or instances. Empirically, we conduct extensive experiments to verify our theories under different attacks on a range of datasets. We show that the global model with a tighter privacy guarantee always provides stronger robustness certification in terms of the *certified attack cost*, while it may exhibit tradeoffs regarding the *certified prediction*. We believe our work will inspire future research of developing certifiably robust DPFL based on its inherent properties.

## 1    INTRODUCTION

Federated Learning (FL), which aims to jointly train a global model with distributed local data, has been widely applied in different applications, such as finance (Yang et al., 2019b), medical analysis (Brisimi et al., 2018), and user behavior prediction (Hard et al., 2018; Yang et al., 2018; 2019a). However, the fact that the local data and the training process are entirely controlled by the *local users* who may be adversarial raises great concerns from both security and privacy perspectives. In particular, recent studies show that FL is vulnerable to different types of training-time attacks, such as model poisoning (Bhagoji et al., 2019), backdoor attacks (Bagdasaryan et al., 2020; Xie et al., 2019; Wang et al., 2020), and label-flipping attacks (Fung et al., 2020). Further, privacy concerns have motivated the need to keep the raw data on local devices without sharing. However, sharing other indirect information such as gradients or model updates as part of the FL training process can also leak sensitive user information (Zhu et al., 2019; Geiping et al., 2020; Bhowmick et al., 2018; Melis et al., 2019). As a result, approaches based on differential privacy (DP) (Dwork & Roth, 2014), homomorphic encryption (Bost et al., 2015; Rouhani et al., 2018; Gilad-Bachrach et al., 2016), and secure multiparty computation (Ben-Or et al., 1988; Bonawitz et al., 2017) have been proposed to protect privacy of users in federated learning. In particular, differentially private federated learning (DPFL) provides strong information theoretic guarantees on user privacy, while causing relatively low performance overhead (Li et al., 2020b).

Several defenses have been proposed to defend against poisoning attacks in FL. For instance, various robust aggregation methods (Fung et al., 2020; Pillutla et al., 2019; Blanchard et al., 2017; El Mhamdi et al., 2018; Chen et al., 2017b; Yin et al., 2018; Fu et al., 2019; Li et al., 2020a) identify and down-weight the malicious updates during aggregation or estimate a true "center" of the received updates rather than taking a weighted average. Other methods include robust federated training protocols (e.g., clipping (Sun et al., 2019), noisy perturbation (Sun et al., 2019), and additional

evaluation during training (Andreina et al., 2020)) and post-training strategies (e.g., fine-tuning and pruning (Wu et al., 2020)) that repair the poisoned global model. However, as these works mainly focus on providing empirical robustness for FL, they have been shown to be vulnerable to newly proposed strong adaptive attacks (Wang et al., 2020; Xie et al., 2019; Baruch et al., 2019; Fang et al., 2020). Hence, in this paper, we aim to *develop certified robustness guarantees for FL against different poisoning attacks.* Further, as differentially private federated learning (DPFL) is often used to protect user privacy, we also aim to ask: *Can we leverage the innate privacy property of DPFL to provide robustness certification against poisoning attacks for free? Can we further improve the privacy of FL so as to improve its certified robustness?*

Recent studies suggest that differential privacy (DP) is inherently related with robustness of ML models. Intuitively, DP is designed to protect the privacy of individual data, such that the output of an algorithm remains essentially unchanged when one individual input point is modified. Hence, the prediction of a DP model will be less impacted by a small amount of poisoned training data. Consequently, DP has been used to provide both theoretical and empirical defenses against evasion attacks (Lecuyer et al., 2019a) and data poisoning attacks (Ma et al., 2019; Hong et al., 2020) on *centralized* ML models. It has also been used as an empirical defense against backdoor attacks (Gu et al., 2019) in federated learning (Bagdasaryan et al., 2020; Sun et al., 2019), although no theoretical guarantee is provided. To the best of our knowledge, despite of the wide application of DPFL, there is no work providing certified robustness for DPFL leveraging its privacy property.

In this paper, we aim to leverage the inherent privacy property of DPFL to provide robustness certification for FL against poisoning attacks *for free*. Our challenges include: (1) performing privacy analysis over training rounds in DPFL algorithms and (2) theoretically guaranteeing certified robustness based on DP properties under a given privacy budget. We propose two robustness certification criteria for FL: *certified prediction* and *certified attack cost* under different attack constraints. We consider both user-level DP (Agarwal et al., 2018; Geyer et al., 2017; McMahan et al., 2018; Asoodeh & Calmon, 2020; Liang et al., 2020) which is widely guaranteed in FL, and instance-level DP (Malekzadeh et al., 2021; Zhu et al., 2021) which is less explored in FL. We prove that a FL model satisfying user-level DP is certifiably robust against a bounded number of adversarial users. In addition, we propose `InsDP-FedAvg` algorithm to improve instance-level DP in FL, and prove that instance-level DPFL is certifiably robust against a bounded number of adversarial instances. We also study the correlation between privacy guarantee and certified robustness of FL. While stronger privacy guarantees result in greater attack cost, overly strong privacy can hurt the certified prediction by introducing too much noise in the training process. Thus, the optimal certified prediction is often achieved under a proper balance between privacy protection and utility loss.

**Key Contributions.** Our work takes the first step to provide certified robustness in DPFL for free against poisoning attacks. We make contributions on both theoretical and empirical fronts.

- We propose two criteria for certified robustness of FL against poisoning attacks (Section 4.2).
- Given a FL model satisfying user-level DP, we prove that it is certifiably robust against arbitrary poisoning attacks with a bounded number of adversarial users (Section 4.2).
- We propose `InsDP-FedAvg` algorithm to improve FL instance-level privacy guarantee (Section 5.1). We prove that instance-level DPFL is certifiably robust against the manipulation of a bounded number of instances during training (Section 5.2).
- We conduct extensive experiments on image classification of MNIST, CIFAR-10 and sentiment analysis of tweets to verify our proposed certifications of two robustness criteria, and compare the certified results of different DPFL algorithms (Section 6).

## 2 RELATED WORK

**Differentially Private Federated Learning.** Different approaches are proposed to guarantee the user-level privacy for FL. (Geyer et al., 2017; McMahan et al., 2018) clip the norm of each local update, add Gaussian noise on the summed update, and characterize its privacy budget via moment accountant (Abadi et al., 2016). (McMahan et al., 2018) extends (Geyer et al., 2017) to language models. In CpSGD (Agarwal et al., 2018), each user clips and quantizes the model update, and adds noise drawn from Binomial distribution, achieving both communication efficiency and DP. (Bhowmick et al., 2018) derive DP for FL via Rényi divergence (Mironov, 2017) and study its protection against data reconstruction attacks. (Liang et al., 2020) utilizes Laplacian smoothing for each local update to enhance the model utility. Instead of using moment accountant to track privacy

budget over FL rounds as previous work, (Asoodeh & Calmon, 2020) derives the DP parameters by interpreting each round as a Markov kernel and quantify its impact on privacy parameters. All these works only focus on providing *user-level* privacy, leaving its robustness property unexplored.

In terms of instance-level privacy for FL, there are only a few work (Malekzadeh et al., 2021; Zhu et al., 2021). Dopamine (Malekzadeh et al., 2021) provides instance-level privacy guarantee when each user only performs one step of DP-SGD (Abadi et al., 2016) at each FL round. However, it cannot be applied to multi-step SGD for each user, thus it cannot be extended to the general FL setting FedAvg (McMahan et al., 2017). (Zhu et al., 2021) privately aggregate the labels from users in a voting scheme, and provide DP guarantees on both user level and instance level. However, it is also not applicable to standard FL, since it does not allow aggregating the gradients or updates.

**Differential Privacy and Robustness.** In standard (centralized) learning, Pixel-DP (Lecuyer et al., 2019a) is proposed to certify the model robsutness against *evasion* attacks. However, it is unclear how to leverage it to certify against poisoning attacks. To certify the robustness against *poisoning* attacks, (Ma et al., 2019) show that private learners are resistant to data poisoning and analyze the lower bound of attack cost against poisoning attacks for regression models. Here we certify the robustness in DPFL setting with such lower bound as one of our certification criteria and additionally derive its upper bounds. (Hong et al., 2020) show that the off-the-shelf mechanism DP-SGD (Abadi et al., 2016), which clips per-sample gradients and add Guassian noises during training, can serve as a defense against poisoning attacks empirically. In *federated learning*, empirical work (Bagdasaryan et al., 2020; Sun et al., 2019) show that DPFL can mitigate backdoor attacks; however, none of these work provides certified robustness guarantees for DPFL against poisoning attacks.

## 3 PRELIMINARIES

We start by providing some background on differential privacy (DP) and federated learning (FL).

**Differential Privacy (DP).** DP is a formal, mathematically rigorous definition (and standard) of privacy that intuitively guarantees that a randomized algorithm behaves similarly on similar inputs and that the output of the algorithm is about the same whether or not an individual's data is included as part of the input (Dwork & Roth, 2014).

**Definition 1** (($\epsilon, \delta$)-DP (Dwork et al., 2006)). *A randomized mechanism $\mathcal{M} : \mathcal{D} \rightarrow \Theta$ with domain $\mathcal{D}$ and range $\Theta$ satisfies ($\epsilon, \delta$)-DP if for any pair of two adjacent datasets $d, d' \in \mathcal{D}$, and for any possible (measurable) output set $E \subseteq \Theta$, it holds that $\Pr[\mathcal{M}(d) \in E] \leq e^\epsilon \Pr[\mathcal{M}(d') \in E] + \delta$.*

In Definition 1, when $\mathcal{M}$ is a training algorithm for ML model, domain $\mathcal{D}$ and range $\Theta$ represent all possible training datasets and all possible trained models respectively. Group DP for ($\epsilon, \delta$)-DP mechanisms follows immediately from Definition 1 where the privacy guarantee drops with the size of the group. Formally, it says:

**Lemma 1** (Group DP). *For mechanism $\mathcal{M}$ that satisfies ($\epsilon, \delta$)-DP, it satisfies $(k\epsilon, \frac{1-e^{k\epsilon}}{1-e^\epsilon}\delta)$-DP for groups of size $k$. That is, for any $d, d' \in \mathcal{D}$ that differ by $k$ individuals, and any $E \subseteq \Theta$ it holds that $\Pr[\mathcal{M}(d) \in E] \leq e^{k\epsilon} \Pr[\mathcal{M}(d') \in E] + \frac{1-e^{k\epsilon}}{1-e^\epsilon}\delta$.*

**Federated Learning.** FedAvg was introduced by (McMahan et al., 2017) for FL to train a shared global model without direct access to training data of users. Specifically, given a FL system with $N$ users, at round $t$, the server sends the current global model $w_{t-1}$ to users in the selected user set $U_t$, where $|U_t| = m = qN$ and $q$ is the user sampling probability. Each selected user $i \in U_t$ locally updates the model for $E$ local epochs with its dataset $D_i$ and learning rate $\eta$ to obtain a new local model. Then, the user sends the local model updates $\Delta w_t^i$ to the server. Finally, the server aggregates over the updates from all selected users into the new global model $w_t = w_{t-1} + \frac{1}{m}\sum_{i \in U_t} \Delta w_t^i$.

## 4 USER-LEVEL PRIVACY AND CERTIFIED ROBUSTNESS FOR FL

### 4.1 USER-LEVEL PRIVACY AND BACKGROUND

Definition 1 leaves the definition of adjacent datasets flexible, which depends on applications. To protect user-level privacy, adjacent datasets are defined as those differing by data from one user (McMahan et al., 2018). The formal definition of User-level ($\epsilon, \delta$)-DP (Definition 2) is omitted to Appendix A.1.

Following standard DPFL (Geyer et al., 2017; McMahan et al., 2018), we introduce one of standard user-level DPFL algorithms `UserDP-FedAvg` (Algorithm 1 in Appendix A.1). At each round,

the server first clips the update from each user with a threshold $S$ such that its $\ell_2$-sensitivity is upper bounded by $S$. Next, the server sums up the updates, adds Gaussian noise sampled from $\mathcal{N}(0, \sigma^2 S^2)$, and takes the average, i.e., $w_t \leftarrow w_{t-1} + \frac{1}{m}\left(\sum_{i \in U_t} \text{Clip}(\Delta w_t^i, S) + \mathcal{N}\left(0, \sigma^2 S^2\right)\right)$. Given the user sampling probability $q$, noise level $\sigma$, FL rounds $T$, and a $\delta > 0$, the privacy analysis of UserDP-FedAvg satisfying $(\epsilon, \delta)$-DP is given by Proposition 1 in Appendix A.1, which is a generalization of (Abadi et al., 2016). The aim of Proposition 1 is to analyze privacy budget $\epsilon$ in FL, which is accumulated as $T$ increases due to the continuous access to training data. Following (Geyer et al., 2017; McMahan et al., 2018), moment accountant (Abadi et al., 2016) is used in the privacy analysis.

### 4.2 CERTIFIED ROBUSTNESS OF USER-LEVEL DPFL AGAINST POISONING ATTACKS

**Threat Model.** We consider the poisoning attacks against FL, where $k$ adversarial users have poisoned instances in local datasets, aiming to fool the trained DPFL global model. Such attacks include *backdoor* attacks (Gu et al., 2019; Chen et al., 2017a) and *label flipping* attacks (Biggio et al., 2012; Huang et al., 2011). The detailed description of these attacks is deferred to Appendix A.2. Note that our robustness certification is attack-agnostic under certain attack constraints (*e.g.*, $k$), and we will verify our certification bounds with different poisoning attacks in Section 6. Next, we propose two criteria for the robustness certification in FL: *certified prediction* and *certified attack cost*.

**Certified Prediction.** Consider the classification task with $C$ classes. We define the classification *scoring function* $f : (\Theta, \mathbb{R}^d) \to \Upsilon^C$ which maps model parameters $\theta \in \Theta$ and an input data $x \in \mathbb{R}^d$ to a confidence vector $f(\theta, x)$, and $f_c(\theta, x) \in [0, 1]$ represents the confidence of class $c$. We mainly focus on the confidence after normalization, i.e., $f(\theta, x) \in \Upsilon^C = \{p \in \mathbb{R}^C_{\geq 0} : \|p\|_1 = 1\}$ in the probability simplex. Since the DP mechanism $\mathcal{M}$ is randomized and produces a *stochastic* FL global model $\theta = \mathcal{M}(D)$, it is natural to resort to a probabilistic expression as a bridge for quantitative robustness certifications. Following the convention in (Lecuyer et al., 2019b; Ma et al., 2019), we use the expectation of the model's prediction to provide a quantitative guarantee on the robustness of $\mathcal{M}$. Specifically, we define the *expected scoring function* $F : (\theta, \mathbb{R}^d) \to \Upsilon^C$ where $F_c(\mathcal{M}(D), x) = \mathbb{E}[f_c(\mathcal{M}(D), x)]$ is the expected confidence for class $c$. The expectation is taken over DP training randomness, e.g., random Gaussian noise and random user subsampling. The corresponding *prediction* $H : (\theta, \mathbb{R}^d) \to [C]$ is defined by $H(\mathcal{M}(D), x) := \arg\max_{c \in [C]} F_c(\mathcal{M}(D), x)$, which is the top-1 class based on the expected prediction confidence. We will prove that such prediction allows robustness certification against poisoning attacks.

Following our threat model above and DPFL training in Algorithm 1, we denote the trained global model exposed to poisoning attacks by $\mathcal{M}(D')$. When $k = 1$, $D$ and $D'$ are user-level adjacent datasets according to Definition 2. Given that mechanism $\mathcal{M}$ satisfies user-level $(\epsilon, \delta)$-DP, based on the innate DP property, the distribution of the stochastic model $\mathcal{M}(D')$ is "close" to the distribution of $\mathcal{M}(D)$. Moreover, according to the *post-processing property* of DP, during testing, given a test sample $x$, we would expect the values of the expected confidence for each class $c$, i.e., $F_c(\mathcal{M}(D'), x)$ and $F_c(\mathcal{M}(D), x)$, to be close, and hence the returned most likely class to be the *same*, i.e., $H(\mathcal{M}(D), x) = H(\mathcal{M}(D'), x)$, indicating *robust* prediction against poisoning attacks.

**Theorem 1** (Condition for Certified Prediction under One Adversarial User). *Suppose a randomized mechanism $\mathcal{M}$ satisfies user-level $(\epsilon, \delta)$-DP. For two user sets $B$ and $B'$ that differ by one user, let $D$ and $D'$ be the corresponding training datasets. For a test input $x$, suppose $\mathbb{A}, \mathbb{B} \in [C]$ satisfy $\mathbb{A} = \arg\max_{c \in [C]} F_c(\mathcal{M}(D), x)$ and $\mathbb{B} = \arg\max_{c \in [C]: c \neq \mathbb{A}} F_c(\mathcal{M}(D), x)$, then if*

$$F_{\mathbb{A}}(\mathcal{M}(D), x) > e^{2\epsilon} F_{\mathbb{B}}(\mathcal{M}(D), x) + (1 + e^{\epsilon})\delta, \tag{1}$$

*it is guaranteed that $H(\mathcal{M}(D'), x) = H(\mathcal{M}(D), x) = \mathbb{A}$.*

When $k > 1$, we resort to group DP. According to Lemma 1, given mechanism $\mathcal{M}$ satisfying user-level $(\epsilon, \delta)$-DP, it also satisfies user-level $(k\epsilon, \frac{1 - e^{k\epsilon}}{1 - e^{\epsilon}}\delta)$-DP for groups of size $k$. When $k$ is smaller than a certain threshold, leveraging the group DP property, we would expect that the distribution of the stochastic model $\mathcal{M}(D')$ is not too far away from the distribution of $\mathcal{M}(D)$ such that they would make the same prediction for a test sample with probabilistic guarantees. Therefore, the privacy and robustness guarantees are simultaneously met by $\mathcal{M}$.

**Theorem 2** (Upper Bound of $k$ for Certified Prediction). *Suppose a randomized mechanism $\mathcal{M}$ satisfies user-level $(\epsilon, \delta)$-DP. For two user sets $B$ and $B'$ that differ by $k$ users, let $D$ and $D'$ be the corresponding training datasets. For a test input $x$, suppose $\mathbb{A}, \mathbb{B} \in [C]$ satisfy*

$\mathbb{A} = \arg\max_{c \in [C]} F_c(\mathcal{M}(D), x)$ *and* $\mathbb{B} = \arg\max_{c \in [C]: c \neq \mathbb{A}} F_c(\mathcal{M}(D), x)$, *then* $H(\mathcal{M}(D'), x) = H(\mathcal{M}(D), x) = \mathbb{A}$, $\forall k < \mathsf{K}$ *where* $\mathsf{K}$ *is the certified number of adversarial users:*

$$\mathsf{K} = \frac{1}{2\epsilon} \log \frac{F_{\mathbb{A}}(\mathcal{M}(D), x)(e^\epsilon - 1) + \delta}{F_{\mathbb{B}}(\mathcal{M}(D), x)(e^\epsilon - 1) + \delta} \tag{2}$$

The proofs of Theorems 1 and 2 are omitted to Appendix A.4. Theorems 1 and 2 reflect a tradeoff between privacy and certified prediction: (i) in Theorem 1, if $\epsilon$ is large such that the RHS of Eq (1) $> 1$, the robustness condition cannot be met since the expected confidence $F_{\mathbb{A}}(\mathcal{M}(D), x) \in [0, 1]$. However, to achieve small $\epsilon$, i.e., strong privacy, large noise is required during training, which would hurt model utility and thus result in small confidence margin between the top two classes (*e.g.*, $F_{\mathbb{A}}(\mathcal{M}(D), x)$ and $F_{\mathbb{B}}(\mathcal{M}(D), x)$), making it hard to meet the robustness condition. (ii) In Theorem 2 if we fix $F_{\mathbb{A}}(\mathcal{M}(D), x)$ and $F_{\mathbb{B}}(\mathcal{M}(D), x)$, smaller $\epsilon$ of FL can certify larger $\mathsf{K}$. However, smaller $\epsilon$ also induces smaller confidence margin, thus reducing $\mathsf{K}$ instead. As a result, properly choosing $\epsilon$ would help to certify a large $\mathsf{K}$.

**Certified Attack Cost.** In addition to the certified prediction, we define the *attack cost* for attacker $C : \Theta \to \mathbb{R}$ which quantifies the difference between the poisoned model and the *attack goal*. In general, attacker aims to minimize the *expected* attack cost $J(D) := \mathbb{E}[C(\mathcal{M}(D))]$, where the expectation is taken over the randomness of DP training. The cost function can be instantiated according to the concrete attack goal in different types of poisoning attacks, and we provide some examples below. Given a global FL model satisfying user-level $(\epsilon, \delta)$-DP, we will prove the lower bound of the attack cost $J(D')$ when manipulating the data of at most $k$ users. Higher lower bound of the attack cost indicates more *certifiably robust* global model.

**Example 1.** *(Backdoor attack (Gu et al., 2019))* $C(\theta) = \frac{1}{n} \sum_{i=1}^{n} l(\theta, z_i^*)$, *where* $z_i^* = (x_i + \delta_x, y^*)$, $\delta_x$ *is the backdoor pattern,* $y^*$ *is the target adversarial label. Minimizing* $J(D')$ *drives the prediction on any test data with the backdoor pattern* $\delta_x$ *to the target label* $y^*$.

**Example 2.** *(Label Flipping attack (Biggio et al., 2012))* $C(\theta) = \frac{1}{n} \sum_{i=1}^{n} l(\theta, z_i^*)$, *where* $z_i^* = (x_i, y^*)$ *and* $y^*$ *is the target adversarial label. Minimizing* $J(D')$ *drives the prediction on test data* $x_i$ *to the target label* $y^*$.

**Example 3.** *(Parameter-Targeting attack (Ma et al., 2019))* $C(\theta) = \frac{1}{2}\|\theta - \theta^\star\|^2$, *where* $\theta^\star$ *is the target model. Minimizing* $J(D')$ *drives the poisoned model to be close to the target model.*

**Theorem 3** (Attack Cost with $k$ Attackers)**.** *Suppose a randomized mechanism* $\mathcal{M}$ *satisfies user-level* $(\epsilon, \delta)$-*DP. For two user sets* $B$ *and* $B'$ *that differ* $k$ *users,* $D$ *and* $D'$ *are the corresponding training datasets. Let* $J(D)$ *be the expected attack cost where* $|C(\cdot)| \leq \bar{C}$. *Then,*

$$\min\{e^{k\epsilon} J(D) + \frac{e^{k\epsilon} - 1}{e^\epsilon - 1}\delta\bar{C}, \bar{C}\} \geq J(D') \geq \max\{e^{-k\epsilon} J(D) - \frac{1 - e^{-k\epsilon}}{e^\epsilon - 1}\delta\bar{C}, 0\}, \quad \text{if} \quad C(\cdot) \geq 0$$
$$\min\{e^{-k\epsilon} J(D) + \frac{1 - e^{-k\epsilon}}{e^\epsilon - 1}\delta\bar{C}, 0\} \geq J(D') \geq \max\{e^{k\epsilon} J(D) - \frac{e^{k\epsilon} - 1}{e^\epsilon - 1}\delta\bar{C}, -\bar{C}\}, \quad \text{if} \quad C(\cdot) \leq 0 \tag{3}$$

The proof is omitted to Appendix A.4. Theorem 3 provides the upper bounds and lower bounds for attack cost $J(D')$. The lower bounds show that to what extent the attack can reduce $J(D')$ by manipulating up to $k$ users, i.e., how successful the attack can be. The lower bounds depend on the attack cost on clean model $J(D)$, $k$ and $\epsilon$. When $J(D)$ is higher, the DPFL model under poisoning attacks is more robust because the lower bounds are accordingly higher; a tighter privacy guarantee, i.e., smaller $\epsilon$, can also lead to higher robustness certification as it increases the lower bounds; with larger $k$, the attacker ability grows and thus lead to lower possible $J(D')$. The upper bounds show the least adversarial effect brought by $k$ attackers, i.e., how vulnerable the DPFL model is under the optimistic case (*e.g.*, the backdoor pattern is less distinguishable).

Leveraging the lower bounds in Theorem 3, we can lower-bound the minimum number of attackers required to reduce the attack cost to certain level associated with hyperparameter $\tau$ in Corollary 1.

**Corollary 1** (Lower Bound of $k$ Given $\tau$)**.** *Suppose a randomized mechanism* $\mathcal{M}$ *satisfies user-level* $(\epsilon, \delta)$-*DP. Let attack cost function be* $C$, *the expected attack cost be* $J(\cdot)$. *In order to achieve* $J(D') \leq \frac{1}{\tau} J(D)$ *for* $\tau \geq 1$ *when* $0 \leq C(\cdot) \leq \bar{C}$, *or achieve* $J(D') \leq \tau J(D)$ *for* $1 \leq \tau \leq -\frac{\bar{C}}{J(D)}$ *when* $-\bar{C} \leq C(\cdot) \leq 0$, *the number of adversarial users should satisfy:*

$$k \geq \frac{1}{\epsilon} \log \frac{(e^\epsilon - 1) J(D)\tau + \bar{C}\delta\tau}{(e^\epsilon - 1) J(D) + \bar{C}\delta\tau} \quad \text{or} \quad k \geq \frac{1}{\epsilon} \log \frac{(e^\epsilon - 1) J(D)\tau - \bar{C}\delta}{(e^\epsilon - 1) J(D) - \bar{C}\delta} \quad \text{respectively.} \tag{4}$$

The proof is omitted to Appendix A.4. Corollary 1 shows that stronger privacy guarantee (*i.e.*, smaller $\epsilon$) requires more attackers to achieve the same effectiveness of attack, indicating higher robustness.

## 5 INSTANCE-LEVEL PRIVACY AND CERTIFIED ROBUSTNESS FOR FL

### 5.1 INSTANCE-LEVEL PRIVACY

In this section, we introduce the instance-level DP definition, the corresponding algorithm, and the privacy analysis for FL. When DP is used to protect the privacy of individual instance, the trained stochastic FL model should not differ much if one instance is modified. Hence, the adjacent datasets in instance-level DP are defined as those differing by one instance. The formal definition of Instance-level $(\epsilon, \delta)$-DP (Definition 3) is omitted to Appendix A.1.

Dopamine (Malekzadeh et al., 2021) provides the first instance-level privacy guarantee under FedSGD (McMahan et al., 2017). However, it has two limitations. First, its privacy bound is loose. Although FedSGD performs both user and batch sampling during training, Dopamine ignores the privacy gain provided by random user sampling. In this section, we improve the privacy guarantee under FedSGD with privacy amplification via user sampling (Bassily et al., 2014; Abadi et al., 2016). This improvement leads to algorithm `InsDP-FedSGD`, to achieve tighter privacy analysis. We defer the algorithm (Algorithm 2) as well as its privacy guarantee to Appendix A.1.

Besides the loose privacy bound, Dopamine (Malekzadeh et al., 2021) only allows users to perform *one step of DP-SGD* (Abadi et al., 2016) during each FL round. This restriction limits the efficiency of the algorithm and increases the communication overhead. In practice, users in FL are typically allowed to update their local models for many steps before submitting updates to reduce the communication cost. To solve this problem, we further improve `InsDP-FedSGD` to support multiple local steps during each round. Specifically, we propose a novel instance-level DPFL algorithm `InsDP-FedAvg` (Algorithm 3 in Appendix A.1) allowing users to train multiple local SGD steps before submitting the updates. In `InsDP-FedAvg`, each user $i$ performs local DP-SGD so that the local training mechanism $\mathcal{M}^i$ satisfies instance-level DP. Then, the server aggregates the updates. We prove that the global mechanism $\mathcal{M}$ preserves instance-level DP using DP parallel composition theorem (Dwork & Lei, 2009) and moment accountant (Abadi et al., 2016).

Algorithm 3 formally presents the `InsDP-FedAvg` algorithm and the calculation of its privacy budget $\epsilon$. Specifically, at first, local privacy cost $\epsilon_0^i$ is initialized as 0 before FL training. At round $t$, if user $i$ is not selected, its local privacy cost is kept unchanged $\epsilon_t^i \leftarrow \epsilon_{t-1}^i$. Otherwise user $i$ updates local model by running DP-SGD for $V$ local steps with batch sampling probability $p$, noise level $\sigma$ and clipping threshold $S$, and $\epsilon_t^i$ is accumulated upon $\epsilon_{t-1}^i$ via its local moment accountant. Next, the server aggregates the updates from selected users, and leverages $\{\epsilon_t^i\}_{i \in [N]}$ and the parallel composition in Theorem 4 to calculate the global privacy cost $\epsilon_t$. After $T$ rounds, the mechanism $\mathcal{M}$ that outputs the FL global model in Algorithm 3 is instance-level $(\epsilon_T, \delta)$-DP.

**Theorem 4** (`InsDP-FedAvg` Privacy Guarantee)**.** *In Algorithm 3, during round t, if the local mechanism $\mathcal{M}^i$ satisfies $(\epsilon_t^i, \delta)$-DP, then the global mechanism $\mathcal{M}$ satisfies $\left(\max_{i \in [N]} \epsilon_t^i, \delta\right)$-DP.*

The idea behind Theorem 4 is that when $D'$ and $D$ differ in one instance, the modified instance only falls into one local dataset, and thus parallel composition theorem (Dwork & Lei, 2009) can be applied. Then the privacy guarantee corresponds to the worst-case, and is obtained by taking the maximum local privacy cost across all the users. The detailed proof is given in Appendix A.1.

### 5.2 CERTIFIED ROBUSTNESS OF INSTANCE-LEVEL DPFL AGAINST POISONING ATTACKS

**Threat Model.** We consider poisoning attacks under the presence of $k$ poisoned instances. These instances could be controlled by the same or multiple adversarial users. Our robustness certification is agnostic to the attack methods as long as the number of poisoned instances is constrained.

According to the group DP property (Lemma 1) and the post-processing property for FL model with instance-level $(\epsilon, \delta)$-DP, we prove that our robust certification results proposed for user-level DP are also applicable to instance-level DP. Below is the formal theorem (proof is given in Appendix A.4).

**Theorem 5.** *Suppose $D$ and $D'$ differ by $k$ instances, and the randomized mechanism $\mathcal{M}$ satisfies instance-level $(\epsilon, \delta)$-DP. The results in Theorems 1, 2,and 3, and Corollary 1 hold for $\mathcal{M}$, $D$, and $D'$.*

**Comparison with existing certified prediction methods in centralized setting.** The form of Theorem 1 is similar with the robustness condition against test-time attack in Proposition 1 of (Lecuyer et al., 2019a). This is because the derived robustness conditions are both rooted in the DP properties, but ours focus on the robustness against training-time attacks in FL, which is more challenging

considering the distributed nature and the model training dynamics, i.e., the analysis of the privacy budget over training rounds. Our Theorem 1 is also different from previous randomized smoothing-based certifiably robust centralized learning against backdoor (Weber et al., 2020) and label flipping (Rosenfeld et al., 2020). First, our randomness comes from the *inherent training randomness* of user/instance-level $(\epsilon, \delta)$-DP, e.g., user subsampling and Gaussian noise. Thus, the certified robustness *for free* in DPFL means that the *DPFL learning algorithm $\mathcal{M}$ itself is randomized*, and such randomness can lead to the robustness certification with non-trivial quantitative measurement of the randomness. On the contrary, robustness in randomized smoothing-based methods comes from *explicitly* making the *classification process* randomized via adding noise in training datasets (Weber et al., 2020; Rosenfeld et al., 2020), or test samples (Lecuyer et al., 2019a; Cohen et al., 2019) which is easier to measure. Second, our Theorem 1, 2 hold no matter how $\epsilon$ is achieved, which means that we can add different types of noise, leverage different subsampling strategies or even different FL training protocols to achieve user/instance-level $\epsilon$. However, in (Weber et al., 2020; Rosenfeld et al., 2020) different certifications require different types of noise (Laplacian, Gaussian, etc.). Additionally, DP is suitable to characterize the robustness against poisoning since DP composition theorems can be leveraged to track privacy cost $\epsilon$, which captures the training dynamics of ML model parameters without additional assumptions. Otherwise one may need to track the deviations of model parameters by analyzing SGD over training, which is theoretically knotty and often requires strong assumptions on Lipschitz continuity, smoothness or convexity for the trained models.

## 6 EXPERIMENTS

We present evaluations for robustness certifications, expecially Thm. 2, 3 and Cor. 1. We find that 1) there is a tradeoff between *certified prediction* and privacy on certain datasets; 2) a tighter privacy guarantee *always* provides stronger certified robustness in terms of the *certified attack cost*; 3) our lower bounds of certified attack cost are generally tight when $k$ is small. When $k$ is large, they are tight under strong attacks (*e.g.*, large local poisoning ratio $\alpha$). Stronger attacks or tighter certification are requried to further tighten the gap between the emprical robustness and theoretical bounds.

**Data and Model.** We evaluate our robustness certification results with three datasets: image classification on MNIST, CIFAR-10 and text sentiment analysis task on tweets from Sentiment140 (Go et al.) (Sent140), which involves classifying Twitter posts as positive or negative. For image datasets, we use corresponding standard CNN architectures in the differential privacy library (opa, 2021) of PyTorch; for Sent140, we use a LSTM classifier. Following previous work on DP ML (Jagielski et al., 2020; Ma et al., 2019) and backdoor attacks (Tran et al., 2018; Weber et al., 2020) which evaluate with two classes, we focus on binary classification for MNIST (digit 0 and 1) and CIFAR-10 (airplane and bird), and defer the 10-class results to Appendix A.3. We train FL model following Algorithm 1 for user-level privacy and Algorithm 3 for instance-level privacy. We refer the readers to Appendix A.3 for details about the datasets, networks, parameter setups.

**Poisoning Attacks.** We evaluate several state-of-the-art poisoning attacks against the proposed `UserDP-FedAvg` and `InsDP-FedAvg`. We first consider backdoor attacks (**BKD**) (Bagdasaryan et al., 2020) and label flipping attacks (**LF**) (Fung et al., 2020). For `InsDP-FedAvg`, we consider the worst case where $k$ backdoored or lable-flipped instances are fallen into the dataset of one user. For `UserDP-FedAvg`, we additionally evaluate distributed backdoor attack (**DBA**) (Xie et al., 2019), which is claimed to be a more stealthy backdoor attack against FL. Moreover, we consider BKD, LF and DBA via **model replacement** approach (Bagdasaryan et al., 2020) where $k$ attackers train the local models using local datasets with $\alpha$ fraction of poisoned instances, and scale the malicious updates with hyperparameter $\gamma$, i.e., $\Delta w_t^i \leftarrow \gamma \Delta w_t^i$, before sending them to the sever. This way, the malicious updates would have a stronger impact on the FL model. Note that even when attackers perform scaling, after server clipping, the sensitivity of updates is still upper-bounded by the clipping threshold $S$. So the privacy guarantee in Proposition 1 still holds under poisoning attacks via model replacement. Detailed attack setups are presented in Appendix A.3.

**Evaluation Metrics and Setup.** We consider two evaluation metrics based on our robustness certification criteria. The first metric is **certified accuracy**, which is the fraction of the test set for which the poisoned FL model makes correct and consistent predictions compared with the clean FL model. Given a test set of size $n$, for $i$-th test sample, the ground truth label is $y_i$, the output prediction is $c_i$, and the certified number of adversarial users/instances is $\mathsf{K}_i$. We calculate the certified accuracy at $k$ as $\frac{1}{n} \sum_{i=1}^{n} \mathbb{1}\{c_i = y_i \text{ and } \mathsf{K}_i \geq k\}$. The second metric is the **lower bound of**

**attack cost** in Theorem 3: $\underline{J(D')} = \max\{e^{-k\epsilon}J(B) - \frac{1-e^{-k\epsilon}}{e^{\epsilon}-1}\delta\bar{C}, 0\}$. We evaluate the tightness of $\underline{J(D')}$ by comparing it with empirical attack cost $J(D')$. To quantify the robustness, we evaluate the expected class confidence $F_c(\mathcal{M}(D), x)$ for class $c$ via Monte-Carlo sampling. We run the private FL algorithms for $M =$1000 times, with class confidence $f_c^s = f_c(\mathcal{M}(D), x)$ for each time. We compute its expectation to estimate $F_c(\mathcal{M}(D), x) \approx \frac{1}{M}\sum_{s=1}^{M} f_c^s$ and use it to evaluate Theorem 2. In addition, we use Hoeffding's inequality (Hoeffding, 1994) to calibrates the empirical estimation with confidence level parameter $\psi$, and results are deferred to Appendix A.3. In terms of the attack cost, we use Example 1, 2 as the definitions of cost function $C$ for backdoor attacks and label flipping attacks respectively. We follow similar protocol to estimate $J(D')$ for Theorem 3 and Corollary 1.

## 6.1 ROBUSTNESS EVALUATION OF USER-LEVEL DPFL

**Certified Prediction.** Figure 1(a)(b) present the user-level certified accuracy under different $\epsilon$ by training DPFL models with different noise scale $\sigma$. The results on Sent140 dataset is presented in Figure 13 of Appendix. A.3.8. We observe that the largest $k$ can be certified when $\epsilon$ is around 0.6298 in MNIST, 0.1451 in CIFAR-10, and 0.2247 in Sent140 which verifies the tradeoff between $\epsilon$ and certified accuracy as we discussed in Section 4.2. Advanced DP protocols that requires less noise while achieving similar level of privacy are favored to improve the privacy, utility, and certified accuracy simultaneously. Furthermore, we compare the certified accuracy of four different user-level DPFL methods (McMahan et al., 2018; Geyer et al., 2017) given the same privacy budget $\epsilon$. As shown in Figure 14 and Figure 15 of Appendix. A.3.9, the models trained by different DPFL algorithms satisfying same $\epsilon$ have different certified robustness results. This is because even under the same $\epsilon$, different DPFL algorithms $\mathcal{M}$ produce trained models $\mathcal{M}(D)$ with different model performance, thus leading to different certified robustness. More discussion could be found in Appendix. A.3.9.

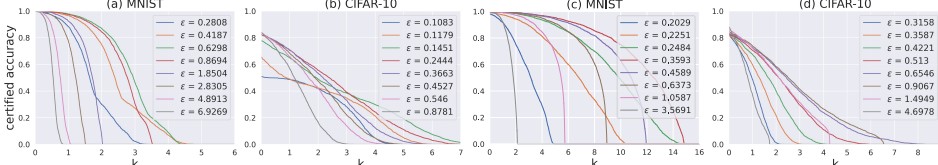

Figure 1: Certified accuracy of FL satisfying user-level DP (a,b), and instance-level DP (c,d).

**Certified Attack Cost.** In order to evaluate Theorem 3 and characterize the tightness of our theoretical lower bound $\underline{J(D')}$, we compare it with the empirical attack cost $J(D')$ under different local poison fraction $\alpha$, attack methods and scale factor $\gamma$ in Figure 2. Note that when $k = 0$, the model is benign so the empirical cost equals to the certified one. We find that 1) when $k$ increases, the attack ability grows, and both the empirical attack cost and theoretical lower bound decreases. 2) In Figure 2 row 1, given the same $k$, higher $\alpha$, i.e., poisoning more local instances for each attacker, achieves a stronger attack, under which lower empirical $J(D)$ can be achieved and is more close to the certified lower bound. This indicates that the lower bound appears tighter when the poisoning attack is stronger. 3) In Figure 2 row 2, we fix $\alpha = 100\%$ and evaluate `UserDP-FedAvg` under different $\gamma$ and attack methods. It turns out that DP serves as a strong defense empirically for FL, given that $J(D)$ did not vary much under different $\gamma(1, 50, 100)$ and different attack methods (BKD, DBA, LF). This is because the clipping operation restricts the magnitude of malicious updates, rendering the model replacement ineffective; the Gaussian noise perturbs the malicious updates and makes the DPFL model stable, and thus the FL model is less likely to memorize the poisoning instances. 4) In both rows, the lower bounds are tight when $k$ is small. When $k$ is large, there remains a gap between our theoretical lower bounds and empirical attack costs under different attacks, which will inspire more effective poisoning attacks or tighter robustness certification.

**Certified Attack Cost under Different $\epsilon$.** Here we further explore the impacts of different factors on the certified attack cost. Figure 3 presents the empirical attack cost and the certified attack cost lower bound given different $\epsilon$ on user-level DP. It is shown that as the privacy guarantee becomes stronger, i.e. smaller $\epsilon$, the model is more robust achieving higher $J(D')$ and $\underline{J(D')}$. In Figure 5 (a)(b), we train user-level $(\epsilon, \delta)$ DPFL models, calculate corresponding $J(D)$, and plot the lower bound of $k$ given different attack effectiveness hyperparameter $\tau$ according to Corollary 1. It shows that 1) when the required attack effectiveness is higher, i.e., $\tau$ is larger, more number of attackers is required. 2) To achieve the same effectiveness of attack, fewer number of attackers is needed for larger $\epsilon$, which means that DPFL model with weaker privacy is more vulnerable to poisoning attacks.

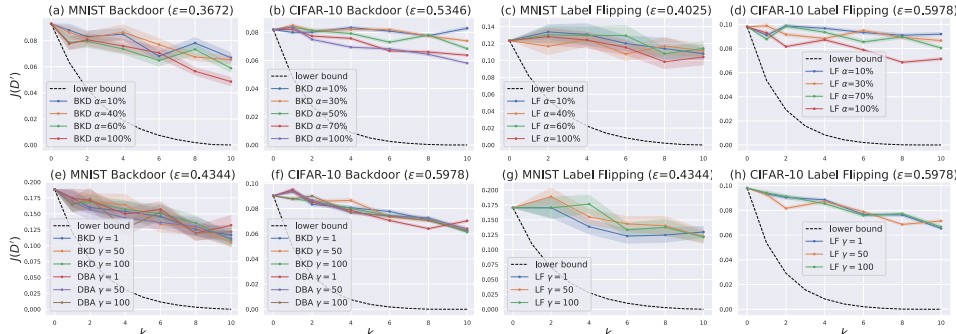

Figure 2: Certified attack cost of user-level DPFL given different $k$, under attacks with different $\alpha$ or $\gamma$.

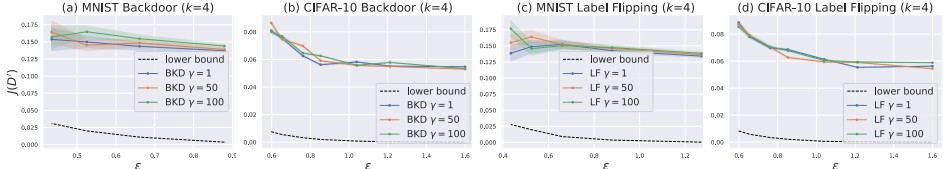

Figure 3: Certified attack cost of user-level DPFL with different $\epsilon$ under different attacks.

## 6.2 ROBUSTNESS EVALUATION OF INSTANCE-LEVEL DPFL

**Certified Prediction.** Figure 1(c)(d) show the instance-level certified accuracy under different $\epsilon$. The optimal $\epsilon$ for K is around $0.3593$ for MNIST and $0.6546$ for CIFAR-10, which is aligned with our observation of the tradeoff between certified accuracy and privacy on user-level DPFL (Section 6.1).

**Certified Attack Cost.** Figure 4 show the certified attack cost on CIFAR-10. From Figure 4 (a)(b), poisoning more instances (i.e., larger $k$) induces lower theoretical and empirical attack cost. From Figure 4 (c)(d), it is clear that instance-level DPFL with stronger privacy guarantee provides higher attack cost both empirically and theoretically, meaning that it is more robust against poisoning attacks. Results on MNIST are deferred to Appendix A.3. Figure 5 (c)(d) show the lower bound of $k$ under different instance-level $\epsilon$ given different $\tau$. Fewer poisoned instances are required to reduce the $J(D')$ to the similar level for a less private DPFL model, indicating that the model is easier to be attacked.

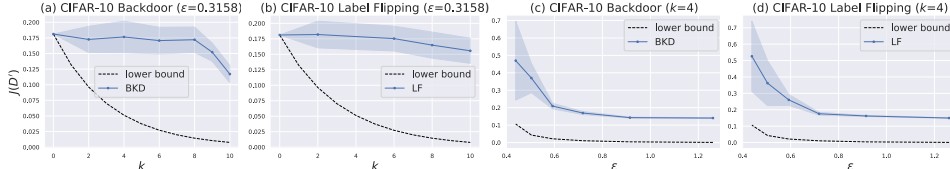

Figure 4: Certified attack cost of instance-level DPFL under different attacks given different number of malicious instances $k$ (a)(b) and different $\epsilon$ (c)(d).

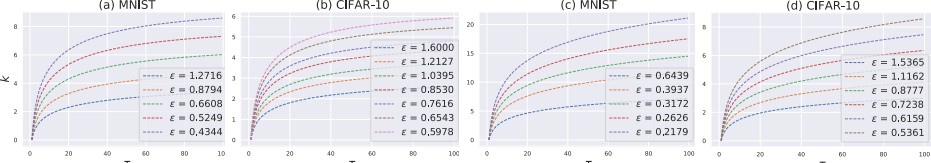

Figure 5: Lower bound of $k$ under user-level $\epsilon$ (a,b) and instance-level $\epsilon$ (c,d) given attack effectiveness $\tau$.

## 7 CONCLUSION

In this paper, we present the *first* work on deriving certified robustness in DPFL for free against poisoning attacks. We propose two robustness certification criteria, based on which we prove that a FL model satisfying user-level (instance-level) DP is certifiably robust against a bounded number of adversarial users (instances). Our theoretical analysis characterizes the inherent relation between certified robustness and differential privacy of FL on both user and instance levels, which are empirically verified with extensive experiments. Our results can be used to improve the trustworthiness of DPFL.

**Ethics Statement.** Our work study the robustness guarantee of differentially private federated learning models from theoretical and empirical perspectives. All the datasets and packages we use are open-sourced. We do not have ethical concerns in our paper.

**Reproducibility Statement.** Our source code is available as the supplemental material for reproducibility purpose. Our experiments can be reproduced following our detailed training and evaluation setups in Appendix A.3. The complete proofs of privacy analysis and certified robustness analysis can be found in the Appendix A.1 and Appendix A.4, respectively.

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

## A APPENDIX

The Appendix is organized as follows:

- Appendix A.1 provides the DP definitions and the DPFL algorithms on both user and instance levels, and the proofs for corresponding privacy guarantees.

- Appendix A.2 specifies our threat models.

- Appendix A.3 provides more details on experimental setups for training and evaluation, the addition experimental results on certified accuracy with confidence level, robustness evaluation of `InsDP-FedAvg` on MNIST, robustness evaluation on 10-class classification, DP bound comparison between `InsDP-FedSGD` and Dopamine, certified accuracy of `UserDP-FedAvg` on Sent140 and certified accuracy comparison of different user-level DPFL algorithms.

- Appendix A.4 provides the proofs for the certified robustness related analysis, including Lemma 1, Theorem 1, 2, 3, 5 and Corollary 1.

- Appendix A.5 provides the comparison to related work (Lecuyer et al., 2019a; Ma et al., 2019).

### A.1 DIFFERENTIALLY PRIVATE FEDERATED LEARNING

#### A.1.1 USERDP-FEDAVG

**Definition 2** (User-level $(\epsilon, \delta)$-DP). *Let $B, B'$ be two user sets with size $N$. Let $D$ and $D'$ be the datasets that are the union of local training examples from all users in $B$ and $B'$ respectively. Then, $D$ and $D'$ are adjacent if $B$ and $B'$ differ by one user. The mechanism $\mathcal{M}$ satisfies user-level $(\epsilon, \delta)$-DP if it meets Definition 1 with $D$ and $D'$ as adjacent datasets.*

---

**Algorithm 1:** `UserDP-FedAvg`.

**Input:** Initial model $w_0$, user sampling probability $q$, privacy parameter $\delta$, clipping threshold $S$, noise level $\sigma$, local datasets $D_1, ..., D_N$, local epochs $E$, learning rate $\eta$.

**Output:** FL model $w_T$ and privacy cost $\epsilon$

**Server executes:**

  **for** *each round $t = 1$ to $T$* **do**

    $m \leftarrow \max(q \cdot N, 1)$;

    $U_t \leftarrow$ (random subset of $m$ users);

    **for** *each user $i \in U_t$ in parallel* **do**

      $\Delta w_t^i \leftarrow$ `UserUpdate`$(i, w_{t-1})$ ;

    $w_t \leftarrow w_{t-1} + \frac{1}{m} \left( \sum_{i \in U_t} \text{Clip}(\Delta w_t^i, S) + \mathcal{N}(0, \sigma^2 S^2) \right)$;

    $\mathcal{M}.accum\_priv\_spending(\sigma, q, \delta)$ ;

$\epsilon = \mathcal{M}.get\_privacy\_spent()$ ;

**return** $w_T, \epsilon$

**Procedure** `UserUpdate`$(i, w_{t-1})$

  $w \leftarrow w_{t-1}$ ;

  **for** *local epoch $e = 1$ to $E$* **do**

    **for** *batch $b \in$ local dataset $D_i$* **do**

      $w \leftarrow w - \eta \nabla l(w; b)$

  $\Delta w_t^i \leftarrow w - w_{t-1}$ ;

  **return** $\Delta w_t^i$

**Procedure** `Clip`$(\Delta, S)$

  **return** $\Delta / \max\left(1, \frac{\|\Delta\|_2}{S}\right)$

---

In Algorithm 1, $\mathcal{M}.accum\_priv\_spending()$ and $\mathcal{M}.get\_privacy\_spent()$ are the calls on the moments accountant $\mathcal{M}$ refer to the API of (Abadi et al., 2016).

Given the user sampling probability $q$, noise level $\sigma$, FL rounds $T$, and a $\delta > 0$, `UserDP-FedAvg` satisfies $(\epsilon, \delta)$-DP as below, which is a generalization of (Abadi et al., 2016). The aim is to analyze privacy budget $\epsilon$, which is accumulated as $T$ increases due to the continuous access to training data.

**Proposition 1** (`UserDP-FedAvg` Privacy Guarantee). *There exist constants $c_1$ and $c_2$ so that given user sampling probability $q$, and FL rounds $T$, for any $\varepsilon < c_1 q^2 T$, if $\sigma \geq c_2 \frac{q\sqrt{T \log(1/\delta)}}{\epsilon}$, the randomized mechanism $\mathcal{M}$ in Algorithm 1 is $(\epsilon, \delta)$-DP for any $\delta > 0$.*

*Proof.* The proof follows the proof of Theorem 1 in (Abadi et al., 2016), while the notations have slightly different meanings under FL settings. In Proposition 1, we use $q$ to represent *user-level* sampling probability and $T$ to represent FL training rounds. □

Note that the above privacy analysis can be further improved by Rényi Differential Privacy (Mironov et al., 2019).

**Discussion** (Li et al., 2020b) divide the user-level privacy into global privacy (Geyer et al., 2017; McMahan et al., 2018) and local privacy (Agarwal et al., 2018). In both local and global privacy, the norm of each update is clipped. The difference lies in that the noise is added on the aggregated model updates in global privacy because a trusted server is assumed, while the noise is added on each local update in local privacy because it assumes that the central server might be malicious. Algorithm 1 belongs to global privacy.

### A.1.2  INSDP-FEDSGD

**Definition 3** (Instance-level $(\epsilon, \delta)$-DP). *Let $D$ be the dataset that is the union of local training examples from all users. Then, $D$ and $D'$ are adjacent if they differ by one instance. The mechanism $\mathcal{M}$ is instance-level $(\epsilon, \delta)$-DP if it meets Definition 1 with $D$ and $D'$ as adjacent datasets.*

---

**Algorithm 2:** `InsDP-FedSGD`.

**Input:** Initial model $w_0$, user sampling probability $q$, privacy parameter $\delta$, local clipping threshold $S$, local noise level $\sigma$, local datasets $D_1, ..., D_N$, learning rate $\eta$, batch sampling probability $p$.

**Output:** FL model $w_T$ and privacy cost $\epsilon$

**Server executes:**
  **for** *each round $t = 1$ to $T$* **do**
    $m \leftarrow \max(q \cdot N, 1)$;
    $U_t \leftarrow$ (random subset of $m$ clients);
    **for** *each user $i \in U_t$ in parallel* **do**
      $\Delta w_t^i \leftarrow$ `UserUpdate`$(i, w_{t-1})$ ;
    $w_t \leftarrow w_{t-1} + \frac{1}{m} \sum_{i \in U_t} \Delta w_t^i$ ;
    $\mathcal{M}.accum\_priv\_spending(\sqrt{m}\sigma, pq, \delta)$
  $\epsilon = \mathcal{M}.get\_privacy\_spent()$ ;
  **return** $w_T, \epsilon$

**Procedure** `UserUpdate`$(i, w_{t-1})$
  $w \leftarrow w_{t-1}$ ;
  $b_t^i \leftarrow$(uniformly sample a batch from $D_i$ with probability $p = L/|D_i|$);
  **for** *each $x_j \in b_t^i$* **do**
    $g(x_j) \leftarrow \nabla l(w; x_j)$;
    $\bar{g}(x_j) \leftarrow$ `Clip`$(g(x_j), S)$ ;
  $\tilde{g} \leftarrow \frac{1}{L} \left( \sum_j \bar{g}(x_j) + \mathcal{N}\left(0, \sigma^2 S^2\right) \right)$;
  $w \leftarrow w - \eta \tilde{g}$ ;
  $\Delta w_t^i \leftarrow w - w_{t-1}$ ;
  **return** $\Delta w_t^i$

**Procedure** `Clip`$(\Delta, S)$
  **return** $\Delta / \max\left(1, \frac{\|\Delta\|_2}{S}\right)$

---

Under FedSGD, when each local model performs one step of DP-SGD (Abadi et al., 2016), the randomized mechanism $\mathcal{M}$ that outputs the global model preserves the instance-level DP. We can regard the one-step update for the global model in Algorithm 2 as:

$$w_t \leftarrow w_{t-1} - \frac{1}{m} \sum_{i \in U_t} \frac{\eta}{L} \left( \sum_{x_j \in b_t^i} \bar{g}(x_j) + \mathcal{N}\left(0, \sigma^2 S^2\right) \right) \tag{5}$$

**Proposition 2** (`InsDP-FedSGD` Privacy Guarantee). *There exist constants $c_1$ and $c_2$ so that given batch sampling probability $p$, and user sampling probability $q$, the number of selected users each round $m$, and FL rounds $T$, for any $\varepsilon < c_1(pq)^2 T$, if $\sigma \geq c_2 \frac{pq\sqrt{T \log(1/\delta)}}{\epsilon \sqrt{m}}$, the randomized mechanism $\mathcal{M}$ in Algorithm 2 is $(\epsilon, \delta)$-DP for any $\delta > 0$.*

*Proof.* i) In instance-level DP, we consider the sampling probability of each instance under the combination of user-level sampling and batch-level sampling. Since the user-level sampling probability is $q$ and the batch-level sampling probablity is $p$, each instance is sampled with probability $pq$. ii) Additionally, since the sensitivity of instance-wise gradient w.r.t one instance is $S$, after local gradient descent and server FL aggregation, the equivalent sensitivity of global model w.r.t one instance is $S' = \frac{\eta S}{Lm}$ according to Eq (5). iii) Moreover, since the local noise is $n_i \sim \mathcal{N}(0, \sigma^2 S^2)$ , then the "virtual" global noise is $n = \frac{\eta}{mL} \sum_{i \in U_t} n_i$ according to Eq (5), so $n \sim \mathcal{N}(0, \frac{\eta^2 \sigma^2 S^2}{mL^2})$. Let $\frac{\eta^2 \sigma^2 S^2}{mL^2} = \sigma'^2 S'^2$ such that $n \sim \mathcal{N}(0, \sigma'^2 S'^2)$. Because $S' = \frac{\eta S}{Lm}$, the equivalent global noise level is $\sigma'^2 = \sigma^2 m$, i.e., $\sigma' = \sigma\sqrt{m}$.

In Proposition 2, we use $pq$ to represent *instance-level* sampling probability, $T$ to represent FL training rounds, $\sigma\sqrt{m}$ to represent the equivalent global noise level. The rest of the proof follows the proof of Theorem 1 in (Abadi et al., 2016).

$\square$

We defer the DP bound evaluation comparison between `InsDP-FedSGD` and Dopamine to Appendix A.3.7.

### A.1.3 INSDP-FEDAVG

---

**Algorithm 3:** `InsDP-FedAvg`.

---

**Input:** Initial model $w_0$, user sampling probability $q$, privacy parameter $\delta$, local clipping threshold $S$, local noise level $\sigma$, local datasets $D_1, ..., D_N$, local steps $V$, learning rate $\eta$, batch sampling probability $p$.

**Output:** FL model $w_T$ and privacy cost $\epsilon$

**Server executes:**

  **for** *each round $t = 1$ to $T$* **do**

    $m \leftarrow \max(q \cdot N, 1)$;

    $U_t \leftarrow$ (random subset of $m$ users);

    **for** *each user $i \in U_t$ in parallel* **do**

      $\Delta w_t^i, \epsilon_t^i \leftarrow$ `UserUpdate`$(i, w_{t-1})$ ;

    **for** *each user $i \notin U_t$* **do**

      $\epsilon_t^i \leftarrow \epsilon_{t-1}^i$ ;

    $w_t \leftarrow w_{t-1} + \frac{1}{m} \sum_{i \in U_t} \Delta w_t^i$ ;

    $\epsilon_t = \mathcal{M}.parallel\_composition(\{\epsilon_t^i\}_{i \in [N]})$

  $\epsilon = \epsilon_T$ ;

**return** $w_T, \epsilon$

**Procedure** `UserUpdate`$(i, w_{t-1})$

  $w \leftarrow w_{t-1}$ ;

  **for** *each local step $v = 1$ to $V$* **do**

    $b \leftarrow$(uniformly sample a batch from $D_i$ with probability $p = L/|D_i|$);

    **for** *each $x_j \in b$* **do**

      $g(x_j) \leftarrow \nabla l(w; x_j)$;

      $\bar{g}(x_j) \leftarrow$ `Clip`$(g(x_j), S)$ ;

    $\widetilde{g} \leftarrow \frac{1}{L}(\sum_j \bar{g}(x_j) + \mathcal{N}(0, \sigma^2 S^2))$;

    $w \leftarrow w - \eta \widetilde{g}$ ;

    $\mathcal{M}^i.accum\_priv\_spending(\sigma, p, \delta)$ ;

  $\epsilon_t^i = \mathcal{M}^i.get\_privacy\_spent()$ ;

  $\Delta w_t^i \leftarrow w - w_{t-1}$ ;

  **return** $\Delta w_t^i, \epsilon_t^i$

**Procedure** `Clip`$(\Delta, S)$

  **return** $\Delta / \max\left(1, \frac{\|\Delta\|_2}{S}\right)$

---

**Lemma 2** (`InsDP-FedAvg` Privacy Guarantee when $T = 1$). *In Algorithm 3, when $T = 1$, suppose local mechanism $\mathcal{M}^i$ satisfies $(\epsilon^i, \delta)$-DP, then global mechanism $\mathcal{M}$ satisfies $(\max_{i \in [N]} \epsilon^i, \delta)$-DP.*

*Proof.* We can regard federated learning as partitioning a dataset $D$ into $N$ disjoint subsets $\{D_1, D_2, \ldots, D_N\}$. $N$ mechanisms $\{\mathcal{M}^1, \ldots, \mathcal{M}^N\}$ are operated on these $N$ parts separately and each $\mathcal{M}^i$ satisfies its own $\epsilon^i$-DP for $i \in [1, N]$. Note that if $i$-th user is not selected , $\epsilon^i = 0$ because local dataset $D_i$ is not accessed and there is no privacy cost. Without loss of generality, we assume the modified data sample $x'$ ($x \to x'$ causes $D \to D'$) is in the local dataset of $k$-th client $D_k$. Let $D, D'$ be two neighboring datasets ($D_k, D_k'$ are also two neighboring datasets). $\mathcal{M}$ is randomized mechanism that outputs the global model, and $\mathcal{M}^i$ is the randomized mechanism that outputs the local model update $\Delta w^i$. Suppose $w_0$ is the initialized and deterministic global model, and $\{z_1, \ldots, z_N\}$ are randomized local updates. We have a sequence of computations $\{z_1 = \mathcal{M}^1(D_1), z_2 = \mathcal{M}^2(D_2; z_1), z_3 = \mathcal{M}^3(D_3; z_1, z_2) \ldots\}$ and $z = \mathcal{M}(D) = w_0 + \sum_{i=1}^{N} z_i$. Note that if $i$-th user is not selected , $z_i = 0$. According to the parallel composition (Tu), we have

$$\Pr[\mathcal{M}(D) = z]$$

$$= \Pr[\mathcal{M}^1(D_1) = z_1] \Pr[\mathcal{M}^2(D_2; z_1) = z_2] \ldots \Pr[\mathcal{M}^N(D_N; z_1, \ldots, z_{N-1}) = z_N]$$

$$\leq \exp(\epsilon^k) \Pr[\mathcal{M}^k(D_k'; z_1, \ldots, z_{k-1}) = z_k] \prod_{i \neq k} \Pr[\mathcal{M}^i(D_i; z_1, \ldots, z_{i-1}) = z_i]$$

$$= \exp(\epsilon^k) \Pr[\mathcal{M}(D') = z]$$

So $\mathcal{M}$ satisfies $\epsilon^k$-DP when the modified data sample lies in the subset $D_k$. Consider the worst case of where the modified data sample could fall in, we know that $\mathcal{M}$ satisfies $(\max_{i \in [N]} \epsilon^i)$-DP. $\square$

We recall Theorem 4.

**Theorem 4** (`InsDP-FedAvg` Privacy Guarantee). *In Algorithm 3, during round $t$, if the local mechanism $\mathcal{M}^i$ satisfies $(\epsilon_t^i, \delta)$-DP, then the global mechanism $\mathcal{M}$ satisfies $\left(\max_{i \in [N]} \epsilon_t^i, \delta\right)$-DP.*

*Proof.* Again, without loss of generality, we assume the modified data sample $x'$ ($x \to x'$ causes $D \to D'$) is in the local dataset of $k$-th user $D_k$. We first consider the case when all users are selected. At each round $t$, $N$ mechanisms are operated on $N$ disjoint parts and each $\mathcal{M}_t^i$ satisfies

own $\epsilon^i$-DP where $\epsilon^i$ is the privacy cost for accessing the local dataset $D_i$ *for one round* (not accumulating over previous rounds). Let $D, D'$ be two neighboring datasets ($D_k, D'_k$ are also two neighboring datasets). Suppose $z_0 = \mathcal{M}_{t-1}(D)$ is the aggregated randomized global model at round $t-1$, and $\{z_1, \ldots, z_N\}$ are the randomized local updates at round $t$, we have a sequence of computations $\{z_1 = \mathcal{M}_t^1(D_1; z_0), z_2 = \mathcal{M}_t^2(D_2; z_0, z_1), z_3 = \mathcal{M}_t^3(D_3; z_0, z_1, z_2) \ldots\}$ and $z = \mathcal{M}_t(D) = z_0 + \sum_i^N z_i$. We first consider the sequential composition (Dwork & Roth, 2014) to accumulate the privacy cost over FL rounds. According to parallel composition, we have

$$\Pr[\mathcal{M}_t(D) = z]$$

$$= \Pr[\mathcal{M}_{t-1}(D) = z_0] \prod_{i=1}^N \Pr[\mathcal{M}_t^i(D_i; z_0, z_1, \ldots, z_{i-1}) = z_i]$$

$$= \Pr[\mathcal{M}_{t-1}(D) = z_0] \Pr[\mathcal{M}_t^k(D_k; z_0, z_1, \ldots, z_{k-1}) = z_k] \prod_{i \neq k} \Pr[\mathcal{M}_t^i(D_i; z_0, z_1, \ldots, z_{i-1}) = z_i]$$

$$\leq \exp(\epsilon_{t-1}) \Pr[\mathcal{M}_{t-1}(D') = z_0] \exp(\epsilon^k) \Pr[\mathcal{M}_t^k(D'_k; z_0, z_1, \ldots, z_{k-1}) = z_k] \prod_{i \neq k} \Pr[\mathcal{M}_t^i(D_i; z_0, z_1, \ldots, z_{i-1}) = z_i]$$

$$= \exp(\epsilon_{t-1} + \epsilon^k) \Pr[\mathcal{M}_t(D') = z]$$

Therefore, $\mathcal{M}_t$ satisfies $\epsilon_t$-DP, where $\epsilon_t = \epsilon_{t-1} + \epsilon^k$. Because the modified data sample always lies in $D_k$ over $t$ rounds and $\epsilon_0 = 0$, we can have $\epsilon_t = t\epsilon^k$, which means that the privacy guarantee of global mechanism $\mathcal{M}_t$ is only determined by the local mechanism of $k$-th user over $t$ rounds.

Moreover, moment accountant (Abadi et al., 2016) is known to reduce the privacy cost from $\mathcal{O}(t)$ to $\mathcal{O}(\sqrt{t})$. We can use the more advanced composition, i.e., moment accountant, instead of the sequential composition, to accumulate the privacy cost for local mechanism $\mathcal{M}^k$ over $t$ FL rounds. In addition, we consider user subsampling. As described in Algorithm 3, if the user $i$ is not selected at round $t$, then its local privacy cost is kept unchanged at this round.

Take the worst case of where $x'$ could lie in, at round $t$, $\mathcal{M}$ satisfies $\epsilon_t$-DP, where $\epsilon_t = \max_{i \in [N]} \epsilon_t^i$, local mechanism $M^i$ satisfies $\epsilon_t^i$-DP, and the local privacy cost $\epsilon_t^i$ is accumulated via local moment accountant in $i$-th user over $t$ rounds.

$\square$

## A.2 THREAT MODELS

We consider targeted poisoning attacks of two types. In *backdoor* attacks (Gu et al., 2019; Chen et al., 2017a), the goal is to embed a backdoor pattern (i.e., a trigger) during training such that any test input with such pattern will be mis-classified as the target. In *label flipping* attacks (Biggio et al., 2012; Huang et al., 2011), the labels of clean training examples from one source class are flipped to the target class while the features of the data are kept unchanged. In FL, the purpose of backdoor attacks is to manipulate local models with backdoored local data, so that the global model would behave normally on untampered data samples while achieving high attack success rate on clean data (Bagdasaryan et al., 2020). Given the same purpose, *distributed backdoor* attack (DBA) (Xie et al., 2019) decomposes the same backdoor pattern to several smaller ones and embeds them to different local training sets for different adversarial users. The goal of label flipping attack against FL is to manipulate local datasets with flipped labels such that the global model will mis-classify the test data in the source class as the target class. The model replacement (Bagdasaryan et al., 2020) is a more powerful approach to perform the above attacks, where the attackers first train the local models using the poisoned datasets and then scale the malicious updates before sending them to the server. This way, the attacker's updates would have a stronger impact on the FL model. We use the model replacement method to perform poisoning attacks and study the effectiveness of DPFL.

For `UserDP-FedAvg`, we consider backdoor, distributed backdoor, and label flipping attacks via the model replacement approach. Next, we formalize the attack process and introduce the notations. Suppose the attacker controls $k$ adversarial users, i.e., there are $k$ attackers out of $N$ users. Let $B$ be the original user set of $N$ benign users, and $B'$ be the user set that contains $k$ attackers. Let $D := \{D_1, D_2, \ldots, D_N\}$ be the union of original benign local datasets across all users. For a data sample $z_j^i := \{x_j^i, y_j^i\}$ in $D_i$, we denote its backdoored version as $z'^i_j := \{x_j^i + \delta_x, y^*\}$, where $\delta_x$

is the backdoor pattern, $y^*$ is the targeted label; the distributed backdoor attack (DBA) version as $z'^i_j := \{x^i_j + \delta^i_x, y^*\}$, where $\delta^i_x$ is the distributed backdoor pattern for attacker $i$; the label-flipped version as $z'^i_j := \{x^i_j, y^*\}$. Note that the composition of all DBA patterns is equivalent to the backdoor pattern, i.e., $\sum_{i=1}^k \delta^i_x = \delta_x$. We assume attacker $i$ has $\alpha_i$ fraction of poisoned samples in its local dataset $D'_i$. Let $D' := \{D'_1, \ldots, D'_{k-1}, D'_k, D_{k+1}, \ldots, D_N\}$ be the union of local datasets when $k$ attackers are present. The adversarial user $i$ performs model replacement by scaling the model update with hyperparameter $\gamma$ before submitting it to the server, i.e., $\Delta w^i_t \leftarrow \gamma \Delta w^i_t$. In our threat model, we consider the attacker that follows our training protocol and has no control over which users are sampled.

For `InsDP-FedAvg`, we consider both backdoor and label flipping attacks. Since distributed backdoor and model replacement attacks are proposed for adversarial users rather than adversarial instances, we do not consider them for instance-level DPFL. There are $k$ backdoored or label-flipped instances $\{z'_1, z'_2, \ldots, z'_k\}$, which could be controlled by same or multiple users. In our threat model, we consider the attacker that follows our training protocol and has no control over which data partition (or batch) is sampled. Note that we do not assume that the adversaries' poisoning data always be sampled. In our algorithms, each batch is randomly subsampled, so the adversaries cannot control if poisoned data are sampled in each step.

### A.3 EXPERIMENTAL DETAILS AND ADDITIONAL RESULTS

#### A.3.1 DATASETS AND MODELS

We evaluate our robustness certification results with two datasets: MNIST (LeCun & Cortes, 2010) and CIFAR-10 (Krizhevsky, 2009). For each dataset, we use corresponding standard CNN architectures in the differential privacy library (opa, 2021) of PyTorch (Paszke et al., 2019).

**MNIST**: We study an image classification problem of handwritten digits in MNIST. It is a dataset of 70000 28x28 pixel images of digits in 10 classes, split into a train set of 60000 images and a test set of 10000 images. Except Section A.3.6, we consider binary classification on classes 0 and 1, making our train set contain 12665 samples, and the test set 2115 samples. The model consists of two Conv-ReLu-MaxPooling layers and two linear layers.

**CIFAR-10**: We study image classification of vehicles and animals in CIFAR-10. This is a harder dataset than MNIST, consisting of 60000 32x32x3 images, split into a train set of 50000 and a test set of 10000. Except Section A.3.6, we consider binary classification on class airplane and bird, making our train set contain 10000 samples, and the test set 2000 samples. The model consists of four Conv-ReLu-AveragePooling layers and one linear layer. When training on CIFAR10, we follow the standard practice for differential privacy (Abadi et al., 2016; Jagielski et al., 2020) and fine-tune a whole model pre-trained non-privately on the more complex CIFAR100, a similarly sized but more complex benchmark dataset. We can achieve reasonable performance on CIFAR-10 datasets by only training (fine-tuning) few rounds.

**Sent140**: We consider a text sentiment analysis task on tweets from Sentiment140 (Go et al.) (Sent140) which involves classifying Twitter posts as positive or negative. We use a two layer LSTM binary classifier containing 256 hidden units with pretrained 300D GloVe embedding (Pennington et al., 2014). Each twitter account corresponds to a device. We use the same network architecture, non-iid dataset partition method, number of selected user per round, learning rate, batch size, etc. as in (Li et al., 2018), which are summarized in Table 1.

#### A.3.2 TRAINING DETAILS

We simulate the federated learning setup by splitting the training datasets for $N$ FL users in an i.i.d manner. FL users run SGD with learning rate $\eta$, momentum 0.9, weight decay 0.0005 to update the local models. The training parameter setups are summarized in Table 1. Following (McMahan et al., 2018) that use $\delta \approx \frac{1}{N^{1.1}}$ as privacy parameter, for `UserDP-FedAvg` we set $\delta = 0.0029$ according to the total number of users, and for `InsDP-FedAvg` we set $\delta = 0.00001$ according the total number of training samples. Next we summarize the privacy guarantees and clean accuracy offered when we study the certified prediction and certified attack cost, which are also the training parameters setups when $k = 0$ in Figure 1, 2, 3, 4, 5, 8.

| Algorithm | Dataset | #training samples | $N$ | $m$ | $E$ | $V$ | batch size | $\eta$ | $S$ | $\delta$ | $\bar{C}$ |
|---|---|---|---|---|---|---|---|---|---|---|---|
| UserDP-FedAvg | MNIST | 12665 | 200 | 20 | 10 | / | 60 | 0.02 | 0.7 | 0.0029 | 0.5 |
| UserDP-FedAvg | CIFAR-10 | 10000 | 200 | 40 | 5 | / | 50 | 0.05 | 1 | 0.0029 | 0.2 |
| UserDP-FedAvg | Sent140 | 40783 | 805 | 10 | 3 | / | 10 | 0.3 | 0.5 | 0.000001 | / |
| InsDP-FedAvg | MNIST | 12665 | 10 | 10 | / | 25 | 50 | 0.02 | 0.7 | 0.00001 | 0.5 |
| InsDP-FedAvg | CIFAR-10 | 10000 | 10 | 10 | / | 100 | 50 | 0.05 | 1 | 0.00001 | 2 |

Table 1: Dataset description and parameters

**User-level DPFL** In order to study the user-level certified prediction under different privacy guarantee, for MNIST, we set $\epsilon$ to be $0.2808, 0.4187, 0.6298, 0.8694, 1.8504, 2.8305, 4.8913, 6.9269$, which are obtained by training UserDP-FedAvg FL model for 3 rounds with noise level $\sigma = 3.0, 2.3, 1.8, 1.5, 1.0, 0.8, 0.6, 0.5$, respectively (Figure 1(a)). For CIFAR-10, we set $\epsilon$ to be $0.1083, 0.1179, 0.1451, 0.2444, 0.3663, 0.4527, 0.5460, 0.8781$, which are obtained by training UserDP-FedAvg FL model for one round with noise level $\sigma = 10.0, 8.0, 6.0, 4.0, 3.0, 2.6, 2.3, 1.7$, respectively (Figure 1(b)). The clean accuracy (average over 1000 runs) of UserDP-FedAvg under non-DP training ($\epsilon = \infty$) and DP training (varying $\epsilon$) on MNIST and CIFAR-10 are reported in Table. 2 and Table. 3 respectively.

| $\sigma$ ‖ | 0 | 0.5 | 0.6 | 0.8 | 1 | 1.5 | 1.8 | 2.3 | 3 |
|---|---|---|---|---|---|---|---|---|---|
| $\epsilon$ ‖ | $\infty$ | 6.9269 | 4.8913 | 2.8305 | 1.8504 | 0.8694 | 0.6298 | 0.4187 | 0.2808 |
| Clean Acc. ‖ | 99.66% | 99.72% | 99.69% | 99.71% | 99.59% | 98.86% | 97.42% | 89.15% | 72.79% |

Table 2: Clean accuracy of UserDP-FedAvg model on MNIST

| $\sigma$ ‖ | 0 | 1.7 | 2.3 | 2.6 | 3 | 4 | 6 | 8 | 10 |
|---|---|---|---|---|---|---|---|---|---|
| $\epsilon$ ‖ | $\infty$ | 0.8781 | 0.546 | 0.4527 | 0.3663 | 0.2444 | 0.1451 | 0.1179 | 0.1083 |
| Clean Acc. ‖ | 81.90% | 81.82% | 80.09% | 79.27% | 77.89% | 73.07% | 64.36% | 57.92% | 54.59% |

Table 3: Clean accuracy of UserDP-FedAvg model on CIFAR-10

To certify the attack cost under different number of adversarial users $k$ (Figure 2), for MNIST, we set the noise level $\sigma$ to be 2.5. When $k = 0$, after training UserDP-FedAvg for $T = 3, 4, 5$ rounds, we obtain FL models with privacy guarantee $\epsilon = 0.3672, 0.4025, 0.4344$ and clean accuracy (average over $M$ runs) $86.69\%, 88.76\%, 88.99\%$. For CIFAR-10, we set the noise level $\sigma$ to be 3.0. After training UserDP-FedAvg for $T = 3, 4$ rounds under $k = 0$, we obtain FL models with privacy guarantee $\epsilon = 0.5346, 0.5978$ and clean accuracy $78.63\%, 78.46\%$.

With the interest of certifying attack cost under different user-level DP guarantee (Figure 3, Figure 5), we explore the empirical attack cost and the certified attack cost lower bound given different $\epsilon$. For MNIST, we set the privacy guarantee $\epsilon$ to be $1.2716, 0.8794, 0.6608, 0.5249, 0.4344$, which are obtained by training UserDP-FedAvg FL models for 5 rounds under noise level $\sigma = 1.3, 1.6, 1.9, 2.2, 2.5$, respectively, and the clean accuracy for the corresponding models are $99.50\%, 99.06\%, 96.52\%, 93.39\%, 88.99\%$. For CIFAR-10, we set the privacy guarantee $\epsilon$ to be $1.600, 1.2127, 1.0395.0.8530, 0.7616, 0.6543, 0.5978$, which are obtained by training UserDP-FedAvg FL models for 4 rounds under noise level $\sigma = 1.5, 1.8, 2.0, 2.3, 2.5, 2.8, 3.0$, respectively, and the clean accuracy for the corresponding models are $85.59\%, 84.52\%, 83.23\%, 81.90\%, 81.27\%, 79.23\%, 78.46\%$.

**Instance-level DPFL** To certify the prediction for instance-level DPFL under different privacy guarantee, for MNIST, we set privacy cost $\epsilon$ to be $0.2029, 0.2251, 0.2484, 0.3593, 0.4589, 0.6373, 1.0587, 3.5691$, which are obtained by training InsDP-FedAvg FL models for 3 rounds with noise level $\sigma = 15, 10, 8, 5, 4, 3, 2, 1$, respectively (Figure 1(c)). For CIFAR-10, we set privacy cost $\epsilon$ to be $0.3158, 0.3587, 0.4221, 0.5130, 0.6546, 0.9067, 1.4949, 4.6978$, which are obtained by training InsDP-FedAvg FL models for one round with noise level $\sigma = 8, 7, 6, 5, 4, 3, 2, 1$, respectively (Figure 1(d)). The clean accuracy (average over 1000 runs) of InsDP-FedAvg under non-DP

training ($\epsilon = \infty$) and DP training (varying $\epsilon$) on MNIST and CIFAR-10 are reported in Table. 4 and Table. 5 respectively.

| $\sigma \parallel$ | 0 | 1 | 2 | 3 | 4 | 5 | 8 | 10 | 15 |
|---|---|---|---|---|---|---|---|---|---|
| $\epsilon \parallel$ | $\infty$ | 3.5691 | 1.0587 | 0.6373 | 0.4589 | 0.3593 | 0.2484 | 0.2251 | 0.2029 |
| Clean Acc. $\parallel$ | 99.85% | 99.73% | 99.73% | 99.70% | 99.65% | 99.57% | 97.99% | 93.30% | 77.12% |

Table 4: Clean accuracy of `InsDP-FedAvg` model on MNIST

| $\sigma \parallel$ | 0 | 1 | 2 | 3 | 4 | 5 | 6 | 7 | 8 |
|---|---|---|---|---|---|---|---|---|---|
| $\epsilon \parallel$ | $\infty$ | 4.6978 | 1.4949 | 0.9067 | 0.6546 | 0.513 | 0.4221 | 0.3587 | 0.3158 |
| Clean Acc. $\parallel$ | 91.15% | 87.91% | 86.02% | 83.85% | 81.43% | 77.59% | 72.69% | 66.47% | 62.26% |

Table 5: Clean accuracy of `InsDP-FedAvg` model on CIFAR-10

With the aim to study certified attack cost under different number of adversarial instances $k$, for MNIST, we set the noise level $\sigma$ to be 10. When $k = 0$, after training `InsDP-FedAvg` for $T = 4, 9$ rounds, we obtain FL models with privacy guarantee $\epsilon = 0.2383, 0.304$ and clean accuracy (average over $M$ runs) 96.40%, 96.93% (Figure 8(a)(b)). For CIFAR-10, we set the noise level $\sigma$ to be 8.0. After training `InsDP-FedAvg` for one round under $k = 0$, we obtain FL models with privacy guarantee $\epsilon = 0.3158$ and clean accuracy 61.78% (Figure 4(a)(b)).

In order to study the empirical attack cost and certified attack cost lower bound under different instance-level DP guarantee, we set the privacy guarantee $\epsilon$ to be $0.5016, 0.311, 0.2646, 0.2318, 0.2202, 0.2096, 0.205$ for MNIST, which are obtained by training `InsDP-FedAvg` FL models for 6 rounds under noise level $\sigma = 5, 8, 10, 13, 15, 18, 20$, respectively, and the clean accuracy for the corresponding models are $99.60\%, 98.81\%, 97.34\%, 92.29\%, 88.01\%, 80.94\%, 79.60\%$ (Figure 8 (c)(d)). For CIFAR-10, we set the privacy guarantee $\epsilon$ to be $1.261, 0.9146, 0.7187, 0.5923, 0.5038, 0.4385$, which are obtained by training `InsDP-FedAvg` FL models for 2 rounds under noise level $\sigma = 3, 4, 5, 6, 7, 8$, respectively, and the clean accuracy for the corresponding models are $84.47\%, 80.99\%, 76.01\%, 68.65\%, 63.07\%, 60.65\%$ (Figure 4 (c)(d)).

With the intention of exploring the upper bound for $k$ given $\tau$ under different instance-level DP guarantee, for MNIST, we set noise level $\sigma$ to be $5, 8, 10, 13, 20$, respectively, to obtain instance-DP FL models after 10 rounds with privacy guarantee $\epsilon = 0.6439, 0.3937, 0.3172, 0.2626, 0.2179$ and clean accuracy $99.58\%, 98.83\%, 97.58\%, 95.23\%, 85.72\%$ (Figure 5(c)). For CIFAR-10, we set noise level $\sigma$ to be $3, 4, 5, 6, 7, 8$ and train `InsDP-FedAvg` for $T = 3$ rounds, to obtain FL models with privacy guarantee $\epsilon = 1.5365, 1.1162, 0.8777, 0.7238, 0.6159, 0.5361$ and clean accuracy $84.34\%, 80.27\%, 74.62\%, 66.94\%, 62.14\%, 59.75\%$ (Figure 5(d)).

### A.3.3 ADDITIONAL IMPLEMENTATION DETAILS

**(Threat Models)** For the attacks against **UserDP-FedAvg**, by default, the local poison fraction $\alpha = 100\%$, and the scale factor $\gamma = 50$. We use same parameters setups for all $k$ attackers. In terms of label flipping attacks, the attackers swap the label of images in source class (digit 1 for MNIST; bird for CIFAR-10) into the target label (digit 0 for MNIST; airplane for CIFAR-10). In terms of backdoor attacks in MNIST and CIFAR-10, the attackers add a backdoor

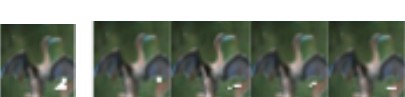

Figure 6: Backdoor pattern (left) and distributed backdoor patterns (right) on CIFAR-10.

pattern, as shown in Figure 6 (left), in images and swap the label of any sample with such pattern into the target label (digit 0 for MNIST; airplane for CIFAR-10). In terms of distributed backdoor attacks, Figure 6 (right) shows an example when the triangle pattern is evenly decomposed into $k = 4$ parts, and they are used as the distributed patterns for $k = 4$ attackers respectively. For the cases where there are more or fewer distributed attackers, the similar decomposition strategy is adopted.

For the attacks against `InsDP-FedAvg`, the same target classes and backdoor patterns are used as `UserDP-FedAvg`. The parameters setups are the same for all $k$ poisoned instances.

(**Robustness Certification**) We certified 2115/2000/1122 test samples from the MNIST/CIFAR-10/Sent140 test sets. In Theorem 3 and Corollary 1 that are related to certified attack cost, $\bar{C}$ specifies the range of $C(\cdot)$. In the implementation, $\bar{C}$ is set to be larger than the maximum empirical attack cost evaluated on the test sets (see Table 1 for details). For each dataset, we use the same $\bar{C}$ for cost function $C$ defined in Example 1 and Example 2. When using Monte-Carlo sampling, we run $M = 1000$ times for certified accuracy, and $M = 100$ times for certified attack cost in all experiments.

(**Machines**) We simulate the federated learning setup (1 server and N users) on a Linux machine with Intel® Xeon® Gold 6132 CPUs and 8 NVidia® 1080Ti GPUs.

(**Libraries**) All code is implemented in Pytorch (Paszke et al., 2019). Please see the submitted code for full details.

### A.3.4 CERTIFIED ACCURACY WITH CONFIDENCE LEVEL

Here we present the certified accuracy with confidence level. We use Hoeffding's inequality (Hoeffding, 1994) to calibrates the empirical estimation with one-sided error tolerance $\psi$, i.e., one-sided confidence level $1 - \psi$. We first use Monte-Carlo sampling by running the private FL algorithms for $M$ times, with class confidence $f_c^s = f_c(\mathcal{M}(D), x)$ for class $c$ each time. We denote the empirical estimation as $\widetilde{F}_c(\mathcal{M}(D), x) = \frac{1}{M} \sum_{s=1}^{M} f_c^s$. For a test input $x$, suppose $\mathbb{A}, \mathbb{B} \in [C]$ satisfy $\mathbb{A} = \arg\max_{c \in [C]} \widetilde{F}_c(\mathcal{M}(D), x)$ and $\mathbb{B} = \arg\max_{c \in [C]: c \neq \mathbb{A}} \widetilde{F}_c(\mathcal{M}(D), x)$. For a given error tolerance $\psi$, we use Hoeffding's inequality to compute a lower bound $\underline{F_{\mathbb{A}}(\mathcal{M}(D), x)}$ on the class confidence $F_{\mathbb{A}}(\mathcal{M}(D), x)$ and a upper bound $\overline{F_{\mathbb{B}}(\mathcal{M}(D), x)}$ on the class confidence $F_{\mathbb{B}}(\mathcal{M}(D), x)$ according to

$$\underline{F_{\mathbb{A}}(\mathcal{M}(D), x)} = \widetilde{F}_{\mathbb{A}}(\mathcal{M}(D), x) - \sqrt{\frac{\log(1/\psi)}{2M}}, \quad \underline{F_{\mathbb{B}}(\mathcal{M}(D), x)} = \widetilde{F}_{\mathbb{B}}(\mathcal{M}(D), x) + \sqrt{\frac{\log(1/\psi)}{2M}}. \tag{6}$$

$\underline{F_{\mathbb{A}}(\mathcal{M}(D), x)}$ and $\overline{F_{\mathbb{B}}(\mathcal{M}(D), x)}$ are used as the expected class confidences for the evaluation of Theorem 2. We use $\psi = 0.01$ and $M = 1000$ for all experiments.

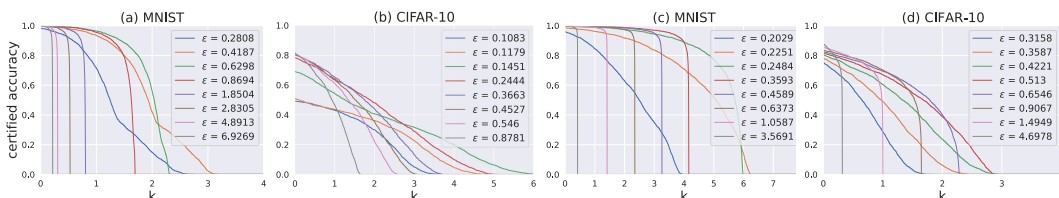

Figure 7: Certified accuracy under 99% confidence of FL satisfying user-level DP (a,b), and instance-level DP (c,d).

As shown in Figure 7, we can observe the same tradeoff between $\epsilon$ and certified accuracy as we discussed in Figure 1. In general, the K in Figure 7 is smaller than the K in Figure 1 because we calibrate the empirical estimation according to Eq. (6), and the class confidence gap between top-1 and top-2 class is narrowed.

### A.3.5 ADDITIONAL ROBUSTNESS EVALUATION OF INSTANCE-LEVEL DPFL

Here we report the robustness evaluation of instance-level DPFL on MNIST. As shown in Figure 8, the results on MNIST are similar to the results on CIFAR-10 in Figure 4.

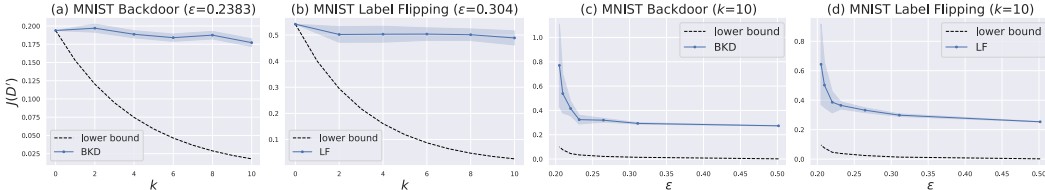

Figure 8: Certified attack cost of instance-level DPFL on MNIST under different attacks given different number of malicious instances $k$ (a)(b) and different $\epsilon$ (c)(d).

### A.3.6 ROBUSTNESS EVALUATION ON 10-CLASS CLASSIFICATION

Here we report the robustness evaluation of user-level DPFL under backdoor attacks on 10-class classification problem. Figure 10 presents the certified accuracy under different $\epsilon$. We can observe the tradeoff between $\epsilon$ and certified accuracy on MNIST. On CIFAR-10, larger $k$ can be certified with smaller $\epsilon$. The certified K is relatively small because we set large $\epsilon$ to preserve a reasonable accuracy for 10-class classification. Our results can inspire advanced DP mechanisms that provide tighter privacy guarantee (i.e., smaller $\epsilon$) while achieving similar level of accuracy. In terms of certified attack cost, as shown in Figure 9 and 11, the trends are similar to the 2-class results in Figure 2, 3 and 5.

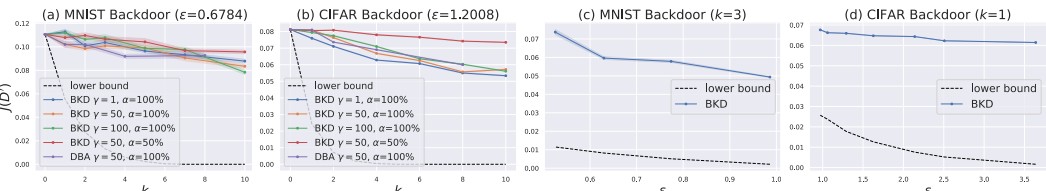

Figure 9: Certified attack cost of user-level DPFL on 10-class classification given different number of malicious instances $k$ (a)(b) and different $\epsilon$ (c)(d).

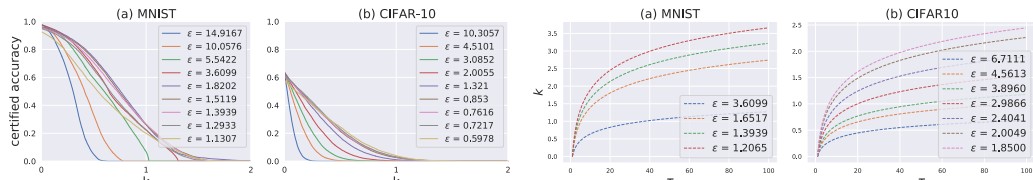

Figure 10: Certified accuracy of FL satisfying user-level DP on 10-class classification.

Figure 11: Lower bound of $k$ on 10-class classification under user-level $\epsilon$ given attack effectiveness $\tau$.

### A.3.7 DP BOUND COMPARISON BETWEEN INSDP-FEDSGD AND DOPAMINE

Here we compare Dopamine to our InsDP-FedSGD, both of which are proposed for FedSGD. Under the same noise level ($\sigma = 3.0$), clipping threshold ($S = 1.5$), user sampling probability ($m/N = 20/30$), and batch sampling probability ($0.4$) settings, both algorithms achieve about 92% accuracy on MNIST (10 classes). The Figure 12 shows the results of privacy guarantee estimation over training rounds, which demonstrates that our method achieves tighter privacy certification. For instance, at round 200, our method ($\epsilon = 1.4029$) achieves a much tighter privacy guarantee than Dopamine ($\epsilon = 2.1303$).

### A.3.8 CERTIFIED ACCURACY OF USERDP-FEDAVG ON SENT140

For Sent140, we set $\epsilon$ to be $0.2238, 0.2247, 0.4102. 0.7382, 1.7151$, which are obtained by training UserDP-FedAvg FL model for three rounds with noise level $\sigma = 4, 3, 2, 1.5, 1$, respectively

As shown in Figure 13, the largest $k$ can be certified when $\epsilon$ is around $0.2247$ in Sent140, which also verifies the tradeoff between $\epsilon$ and certified accuracy as we observed in image datasets.

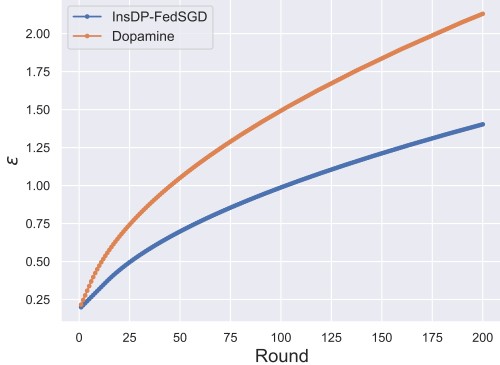

Figure 12: Comparison of DP bound $\epsilon$ under FedSGD on MNIST dataset. Our `InsDP-FedSGD` achieves a tighter DP bound.

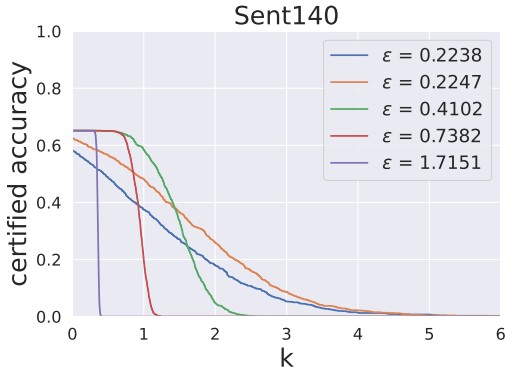

Figure 13: Certified accuracy of FL satisfying user-level DP on Sent140.

### A.3.9 CERTIFIED ACCURACY COMPARISON OF DIFFERENT USER-LEVEL DPFL ALGORITHMS

In this section, we include two more user-level DPFL works (McMahan et al., 2018; Geyer et al., 2017) to study certified accuracy with a total of four different DPFL methods given the same privacy budget $\epsilon$. Since all our proposed robustness certifications are agnostic to DPFL algorithms, i.e., certifications hold no matter how $(\epsilon, \delta)$ is achieved, we can empirically compare the certified results of different DPFL algorithms. Specifically, we consider the following four DPFL algorithms:

- flat clipping (`UserDP-FedAvg`) clips the concatenation of all the layers of model update with the L2 norm threshold $S$.

- per-layer clipping (McMahan et al., 2018) clips each layer of model update with the L2 norm threshold $S$.

- flat median. clipping (Geyer et al., 2017) use the median of norms of clients' model updates as threshold $S$ for flat clipping.

- per-layer median clipping (Geyer et al., 2017) use the median of each layer's norms of clients' model updates as threshold $S$ for per-layer clipping.

For MNIST (CIFAR-10), we set $\epsilon$ to be 0.6319 (0.5346) which is obtained by training all DPFL algorithms with the same noise level $\sigma = 2.3$ ($\sigma = 3.0$) for same number of rounds. For flat clipping and per-layer clipping, we set $S = 0.7$ ($S = 1$) on MNIST (CIFAR-10). Except for local epoch $E = 1$ [1], other FL parameters setups are the same as in Table 1.

---

[1]In experiments we note that the median norm clipping approaches (Geyer et al., 2017) can only be applied when the number of local epoch is small, which makes these methods less practical. Recall that in the server aggregation step, the noise is sampled from $\mathcal{N}(0, \sigma^2 S^2)$, so $S$ cannot be too large in order to keep the amount

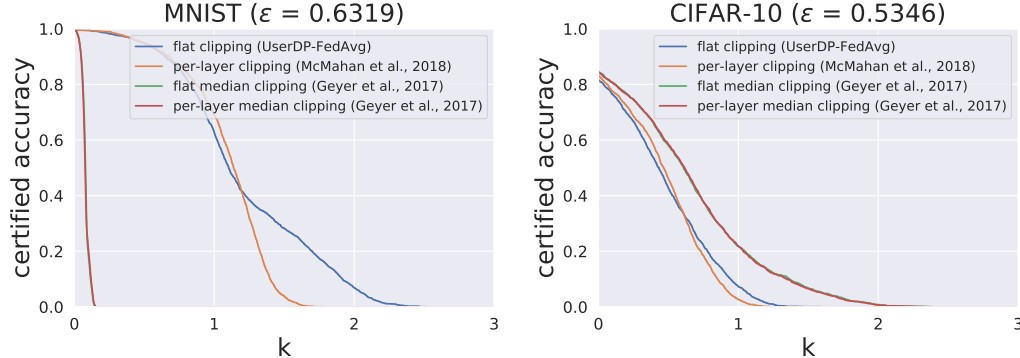

Figure 14: Certified accuracy of model trained by different user-level DPFL algorithms under same $\epsilon$ on MNIST.

Figure 15: Certified accuracy of model trained by different user-level DPFL algorithms under same $\epsilon$ on CIFAR-10.

As shown in Figure 14 and Figure 15, on MNIST, the flat clipping is able to certify the largest number of adversaries $k$; while on CIFAR-10, the median clipping certifies the largest $k$ instead. Moreover, on both MNIST and CIFAR-10, flat clipping and per-layer clipping with the same $S$ lead to different certification results, while the results of flat median clipping and per-layer median clipping are nearly identical. This is because even under the same privacy protection $\epsilon$, different DPFL algorithms $\mathcal{M}$ produce trained models $\mathcal{M}(D)$ with different model performance, thus leading to different certified robustness. Specifically, in Theorem 1, given the same $\epsilon$ and $x$, $F_{\mathbb{A}}(\mathcal{M}(D), x)$ and $F_{\mathbb{B}}(\mathcal{M}(D), x)$ vary for different DPFL trained models $\mathcal{M}(D)$, thus causing different certified K.

The above interesting results indicate that our proposed robustness certifications can serve as new metrics to benchmark the robustness of different DPFL algorithms, which can potentially motivate the investigation for better DPFL algorithms (i.e., different types of noise, clipping methods, subsampling strategies, or even different FL training protocols). We believe these analyses can provide new and important insights to the FL community.

## A.4    PROOFS OF CERTIFIED ROBUSTNESS ANALYSIS

We restate our Lemma 1 here.

**Lemma 1** (Group DP). *For mechanism $\mathcal{M}$ that satisfies $(\epsilon, \delta)$-DP, it satisfies $(k\epsilon, \frac{1-e^{k\epsilon}}{1-e^{\epsilon}}\delta)$-DP for groups of size $k$. That is, for any $d, d' \in \mathcal{D}$ that differ by $k$ individuals, and any $E \subseteq \Theta$ it holds that $\Pr[\mathcal{M}(d) \in E] \leq e^{k\epsilon}\Pr[\mathcal{M}(d') \in E] + \frac{1-e^{k\epsilon}}{1-e^{\epsilon}}\delta$.*

*Proof.* We denote $d$ as $d_0$, $d'$ as $d_k$. $d_i$ differ $i$ individuals with $d_0$. For any $i \in [1, k]$, $d_i$ and $d_{i-1}$ differ by one individual, thus

$$\Pr[M(d_{i-1})] \leq e^{\epsilon}\Pr[M(d_i)] + \delta. \tag{7}$$

By iteratively applying Eq. (7) $k$ times, we have

$$\Pr[M(d_0)] \leq e^{k\epsilon}\Pr[M(d_k)] + (1 + e^{\epsilon} + e^{2\epsilon} + \ldots + e^{(k-1)\epsilon})\delta$$
$$= e^{k\epsilon}\Pr[M(d_k)] + \frac{1-e^{k\epsilon}}{1-e^{\epsilon}}\delta$$

$\square$

Before we prove Theorem 1, we introduce the following lemma:

---

of noise reasonable and preserve good model utility. As more local epoch leads to larger norm of model updates, we set the local epoch as 1 to keep the median norm small.

**Lemma 3.** *Suppose a randomized mechanism $\mathcal{M}$ satisfies user-level $(\epsilon, \delta)$-DP. For two user sets $B$ and $B'$ that differ by one user, $D$ and $D'$ are the corresponding training datasets. For a test input $x$, for any $c \in [C]$, $f_c(\mathcal{M}(D), x) \in [0, 1]$ is the class confidence, then the expected class confidence $F_c(\mathcal{M}(D), x) := \mathbb{E}[f_c(\mathcal{M}(D), x)]$ meets the following property:*

$$F_c(\mathcal{M}(D), x) \leq e^\epsilon F_c(\mathcal{M}(D'), x) + \delta \tag{8}$$

*Proof.* Define $\Theta(a) := \{\theta : f_c(\theta, x) > a\}$. Then

$$\begin{aligned}
F_c(\mathcal{M}(D), x) = \mathbb{E}[f_c(\mathcal{M}(D), x)] &= \int_0^1 \mathbb{P}\left[f_c(\mathcal{M}(D), x) > a\right] da \\
&= \int_0^1 \mathbb{P}\left[\mathcal{M}(D) \in \Theta(a)\right] da \\
&\leq \int_0^1 \left(e^\epsilon \mathbb{P}\left[\mathcal{M}(D') \in \Theta(a)\right] + \delta\right) da \\
&= \int_0^1 e^\epsilon \mathbb{P}\left[f_c(\mathcal{M}(D'), x) > a\right] da + \int_0^1 \delta da \\
&= e^\epsilon F_c(\mathcal{M}(D'), x) + \delta
\end{aligned}$$

$\square$

We recall Theorem 1.

**Theorem 1** (Condition for Certified Prediction under One Adversarial User). *Suppose a randomized mechanism $\mathcal{M}$ satisfies user-level $(\epsilon, \delta)$-DP. For two user sets $B$ and $B'$ that differ by one user, let $D$ and $D'$ be the corresponding training datasets. For a test input $x$, suppose $\mathbb{A}, \mathbb{B} \in [C]$ satisfy $\mathbb{A} = \arg\max_{c \in [C]} F_c(\mathcal{M}(D), x)$ and $\mathbb{B} = \arg\max_{c \in [C]: c \neq \mathbb{A}} F_c(\mathcal{M}(D), x)$, then if*

$$F_\mathbb{A}(\mathcal{M}(D), x) > e^{2\epsilon} F_\mathbb{B}(\mathcal{M}(D), x) + (1 + e^\epsilon)\delta, \tag{1}$$

*it is guaranteed that $H(\mathcal{M}(D'), x) = H(\mathcal{M}(D), x) = \mathbb{A}$.*

*Proof.* According to Lemma 3,

$$F_\mathbb{A}(\mathcal{M}(D), x) \leq e^\epsilon F_\mathbb{A}(\mathcal{M}(D'), x) + \delta \tag{9}$$

$$F_\mathbb{B}(\mathcal{M}(D'), x) \leq e^\epsilon F_\mathbb{B}(\mathcal{M}(D), x) + \delta. \tag{10}$$

Then

$$\begin{aligned}
F_\mathbb{A}(\mathcal{M}(D'), x) &\geq \frac{F_\mathbb{A}(\mathcal{M}(D), x) - \delta}{e^\epsilon} && \text{(Because of Eq. 9)} \\
&\geq \frac{e^{2\epsilon} F_\mathbb{B}(\mathcal{M}(D), x) + (1 + e^\epsilon)\delta - \delta}{e^\epsilon} && \text{(Because of the given condition Eq. 1)} \\
&= e^\epsilon F_\mathbb{B}(\mathcal{M}(D), x) + \delta \\
&\geq e^\epsilon \left(\frac{F_\mathbb{B}(\mathcal{M}(D'), x) - \delta}{e^\epsilon}\right) + \delta && \text{(Because of Eq. 10)} \\
&= F_\mathbb{B}(\mathcal{M}(D'), x),
\end{aligned}$$

which indicates that the prediction of $\mathcal{M}(D')$ at $x$ is $\mathbb{A}$ by definition. $\square$

Before we prove Theorem 2, we introduce the following lemma:

**Lemma 4.** *Suppose a randomized mechanism $\mathcal{M}$ satisfies user-level $(\epsilon, \delta)$-DP. For two user sets $B$ and $B'$ that differ $k$ users, $D$ and $D'$ are the corresponding training datasets. For a test input $x$, for any $c \in [C]$, $f_c(\mathcal{M}(D), x) \in [0, 1]$ is the class confidence, then the expected class confidence $F_c(\mathcal{M}(D), x) := \mathbb{E}[f_c(\mathcal{M}(D), x)]$ meets the following property:*

$$F_c(\mathcal{M}(D), x) \leq e^{k\epsilon} F_c(\mathcal{M}(D'), x) + \frac{1 - e^{k\epsilon}}{1 - e^\epsilon}\delta \tag{11}$$

*Proof.* Define $\Theta(a) := \{\theta : f_c(\theta, x) > a\}$. Then

$$
\begin{aligned}
F_c(\mathcal{M}(D), x) &= \int_0^1 \mathbb{P}\left[f_c(\mathcal{M}(D), x) > a\right] da \\
&= \int_0^1 \mathbb{P}\left[\mathcal{M}(D) \in \Theta(a)\right] da \\
&\le \int_0^1 \left(e^{k\epsilon} \mathbb{P}\left[\mathcal{M}(D') \in \Theta(a)\right] + \frac{1 - e^{k\epsilon}}{1 - e^{\epsilon}} \delta\right) da \\
&\qquad\qquad \text{(Because of Group DP property in Lemma 1)} \\
&= \int_0^1 e^{k\epsilon} \mathbb{P}\left[f_c(\mathcal{M}(D'), x) > a\right] da + \int_0^1 \frac{1 - e^{k\epsilon}}{1 - e^{\epsilon}} \delta da \\
&= e^{k\epsilon} F_c(\mathcal{M}(D'), x) + \frac{1 - e^{k\epsilon}}{1 - e^{\epsilon}} \delta
\end{aligned}
$$

$\square$

We recall Theorem 2.

**Theorem 2** (Upper Bound of $k$ for Certified Prediction). *Suppose a randomized mechanism $\mathcal{M}$ satisfies user-level $(\epsilon, \delta)$-DP. For two user sets $B$ and $B'$ that differ by $k$ users, let $D$ and $D'$ be the corresponding training datasets. For a test input $x$, suppose $\mathbb{A}, \mathbb{B} \in [C]$ satisfy $\mathbb{A} = \arg\max_{c \in [C]} F_c(\mathcal{M}(D), x)$ and $\mathbb{B} = \arg\max_{c \in [C]:c \neq \mathbb{A}} F_c(\mathcal{M}(D), x)$, then $H(\mathcal{M}(D'), x) = H(\mathcal{M}(D), x) = \mathbb{A}, \forall k < \mathsf{K}$ where $\mathsf{K}$ is the certified number of adversarial users:*

$$
\mathsf{K} = \frac{1}{2\epsilon} \log \frac{F_{\mathbb{A}}(\mathcal{M}(D), x)(e^{\epsilon} - 1) + \delta}{F_{\mathbb{B}}(\mathcal{M}(D), x)(e^{\epsilon} - 1) + \delta} \tag{2}
$$

*Proof.* According to Lemma 4, we have

$$
F_{\mathbb{A}}(\mathcal{M}(D), x) \le e^{k\epsilon} F_{\mathbb{A}}(\mathcal{M}(D'), x) + \frac{1 - e^{k\epsilon}}{1 - e^{\epsilon}} \delta \tag{12}
$$

$$
F_{\mathbb{B}}(\mathcal{M}(D'), x) \le e^{k\epsilon} F_{\mathbb{B}}(\mathcal{M}(D), x) + \frac{1 - e^{k\epsilon}}{1 - e^{\epsilon}} \delta. \tag{13}
$$

We can re-write the given condition $k < \mathsf{K}$ according to Eq. (2) as

$$
e^{2k\epsilon} F_{\mathbb{B}}(\mathcal{M}(D), x) + (1 + e^{k\epsilon}) \frac{1 - e^{k\epsilon}}{1 - e^{\epsilon}} \delta < F_{\mathbb{A}}(\mathcal{M}(D), x). \tag{14}
$$

Then

$$
\begin{aligned}
F_{\mathbb{A}}(\mathcal{M}(D'), x) &\ge \frac{F_{\mathbb{A}}(\mathcal{M}(D), x) - \frac{1 - e^{k\epsilon}}{1 - e^{\epsilon}} \delta}{e^{k\epsilon}} \qquad\qquad \text{(Because of Eq. 12)} \\
&> \frac{e^{2k\epsilon} F_{\mathbb{B}}(\mathcal{M}(D), x) + (1 + e^{k\epsilon}) \frac{1 - e^{k\epsilon}}{1 - e^{\epsilon}} \delta - \frac{1 - e^{k\epsilon}}{1 - e^{\epsilon}} \delta}{e^{k\epsilon}} \\
&\qquad\qquad\qquad \text{(Because of the given condition Eq.14)} \\
&= e^{k\epsilon} F_{\mathbb{B}}(\mathcal{M}(D), x) + \frac{1 - e^{k\epsilon}}{1 - e^{\epsilon}} \delta \\
&\ge e^{k\epsilon} \left(\frac{F_{\mathbb{B}}(\mathcal{M}(D'), x) - \frac{1 - e^{k\epsilon}}{1 - e^{\epsilon}} \delta}{e^{k\epsilon}}\right) + \frac{1 - e^{k\epsilon}}{1 - e^{\epsilon}} \delta \qquad \text{(Because of Eq. 13)} \\
&= F_{\mathbb{B}}(\mathcal{M}(D'), x),
\end{aligned}
$$

which indicates that the prediction of $\mathcal{M}(D')$ at $x$ is $\mathbb{A}$ by definition. $\square$

We recall Theorem 3.

**Theorem 3** (Attack Cost with $k$ Attackers). *Suppose a randomized mechanism $\mathcal{M}$ satisfies user-level $(\epsilon, \delta)$-DP. For two user sets $B$ and $B'$ that differ $k$ users, $D$ and $D'$ are the corresponding training datasets. Let $J(D)$ be the expected attack cost where $|C(\cdot)| \leq \bar{C}$. Then,*

$$\min\{e^{k\epsilon}J(D) + \frac{e^{k\epsilon} - 1}{e^{\epsilon} - 1}\delta\bar{C}, \bar{C}\} \geq J(D') \geq \max\{e^{-k\epsilon}J(D) - \frac{1 - e^{-k\epsilon}}{e^{\epsilon} - 1}\delta\bar{C}, 0\}, \quad if \quad C(\cdot) \geq 0$$

$$\min\{e^{-k\epsilon}J(D) + \frac{1 - e^{-k\epsilon}}{e^{\epsilon} - 1}\delta\bar{C}, 0\} \geq J(D') \geq \max\{e^{k\epsilon}J(D) - \frac{e^{k\epsilon} - 1}{e^{\epsilon} - 1}\delta\bar{C}, -\bar{C}\}, \quad if \quad C(\cdot) \leq 0$$

$$(3)$$

*Proof.* We first consider $C(\cdot) \geq 0$. Define $\Theta(a) = \{\theta : C(\theta) > a\}$.

$$J(D) = \int_0^{\bar{C}} \mathbb{P}\left[C(\mathcal{M}(D)) > a\right] da$$

$$= \int_0^{\bar{C}} \mathbb{P}\left[\mathcal{M}(D)) \in \Theta(a)\right] da$$

$$\leq \int_0^{\bar{C}} \left(e^{k\epsilon}\mathbb{P}\left[\mathcal{M}(D')) \in \Theta(a)\right] + \frac{1 - e^{k\epsilon}}{1 - e^{\epsilon}}\delta\right) da$$
$$\text{(Because of Group DP property in Lemma 1)}$$

$$= \int_0^{\bar{C}} e^{k\epsilon}\mathbb{P}\left[\mathcal{M}(D')) \in \Theta(a)\right] da + \frac{1 - e^{k\epsilon}}{1 - e^{\epsilon}}\delta\bar{C}$$

$$= \int_0^{\bar{C}} e^{k\epsilon}\mathbb{P}\left[C(\mathcal{M}(D')) > a\right] da + \frac{1 - e^{k\epsilon}}{1 - e^{\epsilon}}\delta\bar{C}$$

$$= e^{k\epsilon}J(D') + \frac{1 - e^{k\epsilon}}{1 - e^{\epsilon}}\delta\bar{C}$$

i.e.,

$$J(D') \geq e^{-k\epsilon}J(D) - \frac{1 - e^{-k\epsilon}}{e^{\epsilon} - 1}\delta\bar{C}.$$

Switch the role of $D$ and $D'$, we have

$$J(D') \leq e^{k\epsilon}J(D) + \frac{1 - e^{k\epsilon}}{1 - e^{\epsilon}}\delta\bar{C}.$$

Also note that $0 \leq J(D') \leq \bar{C}$ trivially holds due to $0 \leq C(\cdot) \leq \bar{C}$, thus

$$\min\{e^{k\epsilon}J(D) + \frac{e^{k\epsilon} - 1}{e^{\epsilon} - 1}\delta\bar{C}, \bar{C}\} \geq J(D') \geq \max\{e^{-k\epsilon}J(D) - \frac{1 - e^{-k\epsilon}}{e^{\epsilon} - 1}\delta\bar{C}, 0\}.$$

Next we consider $C(\cdot) \leq 0$. Define $\Theta(a) = \{\theta : C(\theta) < a\}$.

$$J(D) = -\int_{-\bar{C}}^0 \mathbb{P}\left[C(\mathcal{M}(D)) < a\right] da$$

$$= -\int_{-\bar{C}}^0 \mathbb{P}\left[\mathcal{M}(D)) \in \Theta(a)\right] da$$

$$\geq -\int_{-\bar{C}}^0 \left(e^{k\epsilon}\mathbb{P}\left[\mathcal{M}(D')) \in \Theta(a)\right] + \frac{1 - e^{k\epsilon}}{1 - e^{\epsilon}}\delta\right) da$$
$$\text{(Because of Group DP property in Lemma 1)}$$

$$= -\int_{-\bar{C}}^0 e^{k\epsilon}\mathbb{P}\left[\mathcal{M}(D')) \in \Theta(a)\right] da - \frac{1 - e^{k\epsilon}}{1 - e^{\epsilon}}\delta\bar{C}$$

$$= -\int_{-\bar{C}}^0 e^{k\epsilon}\mathbb{P}\left[C(\mathcal{M}(D')) < a\right] da - \frac{1 - e^{k\epsilon}}{1 - e^{\epsilon}}\delta\bar{C}$$

$$= e^{k\epsilon}J(D') - \frac{1 - e^{k\epsilon}}{1 - e^{\epsilon}}\delta\bar{C}$$

i.e.,

$$J(D') \leq e^{-k\epsilon} J(D) + \frac{1 - e^{-k\epsilon}}{e^\epsilon - 1} \delta \bar{C}.$$

Switch the role of $D$ and $D'$, we have

$$J(D') \geq e^{k\epsilon} J(D) - \frac{1 - e^{k\epsilon}}{1 - e^\epsilon} \delta \bar{C}.$$

Also note that $-\bar{C} \leq J(D') \leq 0$ trivially holds due to $-\bar{C} \leq C(\cdot) \leq 0$, thus

$$\min\{e^{-k\epsilon} J(D) + \frac{1 - e^{-k\epsilon}}{e^\epsilon - 1} \delta \bar{C}, 0\} \geq J(D') \geq \max\{e^{k\epsilon} J(D) - \frac{e^{k\epsilon} - 1}{e^\epsilon - 1} \delta \bar{C}, -\bar{C}\}$$

$\square$

We recall Corollary 1.

**Corollary 1** (Lower Bound of $k$ Given $\tau$). *Suppose a randomized mechanism $\mathcal{M}$ satisfies user-level $(\epsilon, \delta)$-DP. Let attack cost function be $C$, the expected attack cost be $J(\cdot)$. In order to achieve $J(D') \leq \frac{1}{\tau} J(D)$ for $\tau \geq 1$ when $0 \leq C(\cdot) \leq \bar{C}$, or achieve $J(D') \leq \tau J(D)$ for $1 \leq \tau \leq -\frac{\bar{C}}{J(D)}$ when $-\bar{C} \leq C(\cdot) \leq 0$, the number of adversarial users should satisfy:*

$$k \geq \frac{1}{\epsilon} \log \frac{(e^\epsilon - 1) J(D)\tau + \bar{C}\delta\tau}{(e^\epsilon - 1) J(D) + \bar{C}\delta\tau} \quad or \quad k \geq \frac{1}{\epsilon} \log \frac{(e^\epsilon - 1) J(D)\tau - \bar{C}\delta}{(e^\epsilon - 1) J(D) - \bar{C}\delta} \quad respectively. \quad (4)$$

*Proof.* We first consider $C(\cdot) \geq 0$. According to the lower bound in Theorem 3, when $B'$ and $B$ differ $k$ users, $J(D') \geq e^{-k\epsilon} J(D) - \frac{1 - e^{-k\epsilon}}{e^\epsilon - 1} \delta \bar{C}$. Since we require $J(D') \leq \frac{1}{\tau} J(D)$, then $e^{-k\epsilon} J(D) - \frac{1 - e^{-k\epsilon}}{e^\epsilon - 1} \delta \bar{C} \leq \frac{1}{\tau} J(D)$. Rearranging gives the result.

Next, we consider $C(\cdot) \leq 0$. According to the lower bound in Theorem 3, when $B'$ and $B$ differ $k$ users, $J(D') \geq e^{k\epsilon} J(D) - \frac{e^{k\epsilon} - 1}{e^\epsilon - 1} \delta \bar{C}$. Since we require $J(D') \leq \tau J(D)$, then $e^{k\epsilon} J(D) - \frac{e^{k\epsilon} - 1}{e^\epsilon - 1} \delta \bar{C} \leq \tau J(D)$. Rearranging gives the result.

$\square$

We note that all the above robustness certification related proofs are built upon the user-level $(\epsilon, \delta)$-DP property and the Group DP property. According to Definition 2 and Definition 3, the definition of user-level DP and instance-level DP are both induced from DP (Definition 1) despite the different definitions of adjacent datasets. By applying the definition of instance-level $(\epsilon, \delta)$-DP and following the proof steps of Theorem 1, 2, 3 and Corollary 1, we can derive the similar theoretical conclusions for instance-level DP, leading to Theorem 5 to achieve the certifiably robsut FL for free given the DP property.

### A.5    COMPARISON TO (LECUYER ET AL., 2019A; MA ET AL., 2019)

In this section, we summarize our differences in terms of the relationship between DP and robustness compared to (Lecuyer et al., 2019a; Ma et al., 2019).

(Lecuyer et al., 2019a) work on certified prediction against *test-time* attacks while we study DP against *training-time* poisoning attacks in FL. We would like to emphasize that (Lecuyer et al., 2019a) aim to make the classification process Pixel-DP while the *training algorithm* itself does not satisfy DP, so it cannot directly build the relationship between DP and training-time certified robustness.

- Conceptually, in contrast to the connection between Pixel-DP and certified prediction against adversarial examples, our proposed analysis on the connection between DP and certified robustness against data poisoning is new.

- Technically, Pixel-DP adds noise on test data samples during testing *once* while we add noises in updates or gradient *at every training round*. Although the analysis of Pixel-DP and ours are both rooted in the intuition of DP definition, Pixel-DP requires the classifier to make randomized predictions during testing and *the training algorithm itself does not*

*satisfy DP*. In contrast, our defense against data poisoning requires the *learning algorithm*, rather than the *classification process* as in Pixel-DP, to be randomized. We add noise on the updates (user-level DP) or gradient (instance-level DP) over every training round to make the trained FL model satisfy DP, which requires careful privacy budget analysis of the DPFL model over multiple training rounds considering the distributed nature and the model training dynamics.

- Empirically, we explicitly evaluate the relationship between the privacy protection level $\epsilon$ and the certified robustness based on two robustness criteria on three datasets, indicating the fundamental connections quantitatively. Moreover, we compare the certified robustness of different existing user-level DPFL algorithms.

(Ma et al., 2019) analyze the lower bound of certified cost in centralized DP learning while we use the certified cost as one of our certification criteria for different definitions of DP in FL and additionally derive its upper bounds.

