# OpenReview forum: "Certified Robustness for Free in Differentially Private Federated Learning"
_ICLR.cc/2022/Conference — ICLR 2022 Submitted_

### Official Review · Reviewer_oz1G · 2021-10-17

**Correctness:** 4
**Technical Novelty And Significance:** 2
**Empirical Novelty And Significance:** 2
**Recommendation:** 5
**Confidence:** 3

**Main Review:**

I was very intrigued by the paper at first but ended up having a mixed feeling after reading it.

On the one hand, the results presented are correct, the experiments are solid, and the paper is rather complete from theory to evaluation.

On the other hand, I have several concerns about the paper. First, the main result presented is very similar to [1]. Although the attack scenarios are different, the intuition is almost the same. This raises my concern about the novelty of the work. Second, I do not understand why the authors choose to limit the results to the federated setting. IIUC, the robustness claims should generally hold against data poisoning attacks and there is no specific optimization for federated learning. User-level DP is also not limited to federated learning. I would say FL is more of an application scenario here. Third, if the authors really want to focus on federated learning, then the instance-level DP section does not make sense because all federated protocols should preserve user-level DP in practice.

Overall, I think the paper is not ready for publication for now. I encourage the authors to tweak the presentation carefully to make it coherent.


[1] Lecuyer, Mathias, Vaggelis Atlidakis, Roxana Geambasu, Daniel Hsu, and Suman Jana. "Certified robustness to adversarial examples with differential privacy." In 2019 IEEE Symposium on Security and Privacy (SP), pp. 656-672. IEEE, 2019.

**Summary Of The Paper:**

This paper proves that differential privacy indicates certified robustness against poisoning attacks in federated learning.

**Summary Of The Review:**

I do not recommend for acceptance because of (1) lack of novelty; (2) incoherent presentation.

---

> ### Author Response · Authors · 2021-11-21
> **Official Response to Reviewer oz1G  (1/2)**
>
> We thank the reviewer for the valuable comments and suggestions. We address the comments below.
>
> > Q1: Concern about novelty. The main result presented is very similar to [1]. Although the attack scenarios are different, the intuition is almost the same.
>
> Thanks for the comments.  Although intuitively DP is related to robustness given the model smoothness realization, there is *no existing work* formally proving that DP leads to certified robustness against data poisoning attacks in *FL*. Here we take the ﬁrst step to **formally prove and quantify such connections**.
>
> The comparison to [1] is mainly discussed in Section 2 “Related Work” and Section 5.2 “Comparison with existing certiﬁed prediction methods in centralized setting”. Concretely, [1] works on certified prediction against *test-time* attacks while we study *training-time DP* against poisoning attacks in *FL*. We would like to emphasize that [1] aims to make the *classification process* Pixel-DP while the *training algorithm* itself does not satisfy DP, so it cannot directly build the relationship between DP and training-time certified robustness. Compared to [1], both our conceptual and technical contributions are novel.
> 1) **Conceptually**,  in contrast to the connection between Pixel-DP and certified prediction against adversarial examples, our proposed analysis on the connection between DP and certified robustness against data poisoning is new.  In addition, we propose certified prediction and certified attack cost considering both user and instance levels in FL against poisoning attacks, while [1] only works on certified prediction against evasion attacks in centralized learning.
> 2)  **Technically**, we do not add noise to the *test data samples* as in [1]. Although the analysis of [1] and ours are both rooted from the intuition of DP,  Pixel-DP [1] requires the classiﬁer to make *randomized predictions* during testing and *the training algorithm itself is not DP*; thus it cannot protect the privacy of any training instances or certify the robustness against poisoned training instances.  In contrast, our defense against data poisoning only requires the **learning algorithm**, rather than the **classiﬁcation process** as in [1], to be randomized. We add noise on the model updates (user DP) or sample gradients (ins DP) over every training round to make the trained model satisfy DP, which requires careful privacy budget analysis of the DPFL model over multiple training rounds considering the distributed nature and the model training dynamics.
> 3) **Empirically**, we explicitly evaluate the relationship between the privacy protection level $\epsilon$ and the certified robustness on two image datasets and one NLP dataset (see Appendix A.3.8 for details of the NLP dataset), indicating their fundamental connections quantitatively. Moreover, we compare the certified robustness of different user-level DPFL algorithms (see Appendix A.3.9 for details). We believe this is the first work to provide such insights in FL.
>
> As a summary, given the innate DP property of DPFL learning algorithms, we bridge the **connection between DP and certified robustness against poisoning** attacks and achieve the certified robustness against data poisoning for free in FL which satisfies DP in practice. In addition, in order to **improve the DP guarantee**, in this paper we propose novel randomization mechanisms and analysis for instance-level DP (Algorithm 2, Algorithm 3), which is less explored in FL and we believe these analyses will be of great interest to both the FL and robust ML communities.
>
>
> > Q2: Why the authors choose to limit the results to the federated setting?
>
> Thanks for the comment.
> * We focus on the federated setting because FL naturally requires privacy guarantees, and DP is the most popular notion of privacy. As the FL systems already require and satisfy the DP privacy guarantee, we are able to further derive its fundamental connection to certified robustness quantitatively via two robustness criteria “for free”.
> * Moreover, compared to centralized learning,  the user-level DP definition and the number of attackers that can be certifiably tolerated are unique in FL, and we are interested in evaluating its robustness results.
> * Furthermore, under the FL framework, we are able to conduct experiments and compare the certified robustness of different existing DPFL algorithms. In Appendix A.3.9, we show that the models trained by different user-level DPFL algorithms satisfying the same privacy guarantee $\epsilon$ could give different certified results (we will provide more details in our response to your Q3). These are new and first results in FL research,  which indicate that our proposed robustness certifications can serve as new metrics to benchmark the robustness of different DPFL algorithms.

---

> > ### Author Response · Authors · 2021-11-21
> > **Official Response to Reviewer oz1G (2/2)**
> >
> > > Q3:  The robustness claims should generally hold against data poisoning attacks and there is no specific optimization for federated learning. User-level DP is also not limited to federated learning.
> >
> > We agree with the reviewer’s comment that our certified robustness theorems generally hold for any models satisfying $(\epsilon, \delta)$-DP. However, the DPFL parts are specific to FL. Specifically, **the neighborhood definition (User DP and Ins DP) and corresponding privacy budget $\epsilon$ analysis over multiple training rounds under different DPFL algorithms are unique for the distributed training paradigm FL.** For example, in Thm. 2, we show the upper bound of *attackers that can be tolerated* for Certiﬁed Prediction, where the notion of users/attackers is unique in FL compared to centralized learning; in Thm. 4, we propose the privacy analysis for InsDP-FedAvg (Algorithm 3) via *parallel composition theorem*, which is based on the distributed nature of FL.
> >
> > Furthermore,  **the specific optimization for DPFL algorithms (e.g., different types of noise, clipping methods, subsampling strategies, or even different FL training protocols) will lead to different robustness certification results.**
> > Following the suggestion, we conducted new experiments and found that different DPFL algorithms satisfying the same User-level $(\epsilon, \delta)$-DP privacy protection could have different certified accuracy results in practice. As shown in Fig. 14 and Fig.15 of Appendix A.3.9, on MNIST, the flat clipping is able to certify the largest number of adversaries $k$; while on CIFAR-10, the median clipping certifies the largest $k$ instead. Moreover, on both MNIST and CIFAR-10, flat clipping and per-layer clipping with the same clipping threshold $S$ lead to different certification results, while the results of flat median clipping and per-layer median clipping are nearly identical. This is because even under the same $\epsilon$, different DPFL algorithms $\mathcal{M}$ produce trained models $\mathcal{M}(D)$ with different model performance, thus leading to different certified robustness. Specifically, in Theorem 2, given the same $\epsilon$ and $x$, $F_{\mathbb{A}}(\mathcal{M}(D),x)$ and $F_{\mathbb{B}}(\mathcal{M}(D),x)$ vary for different DPFL trained models $\mathcal{M}(D)$, thus causing different certified $k$.  We believe these analyses can provide new and important insights to the FL community.
> >
> >
> > > Q4: The instance-level DP section does not make sense because all federated protocols should preserve user-level DP in practice.
> >
> > Thanks for the valuable comment. We do believe that the instance-level DP in FL is needed and several studies have discussed the vulnerability of FL without such privacy guarantees.
> > * First,  we note that instance-level DP is an open and ongoing topic in FL (Malekzadeh et al., 2021; Zhu et al., 2021) as discussed in our related work.  We also want to bring more papers (Shokri. Et al. 2015, Hitaj et al. 2017) to your attention that focus on the instance-level DP under collaborative learning/federated learning.
> > * Second, the federated protocols that preserve user-level DP only could still be attacked and leak the privacy of user data samples. Specifically, recent studies have shown that it is possible to recover the batch of training data from the gradients of users in distributed training (Zhu et al. 2019) or from the local updates of users in federated learning (Geiping et al. 2020). Furthermore, as shown in (Zhu et al. 2019), adding noises on gradients before sharing does not prevent the leak. Therefore, the user-level privacy may not be enough to protect the training data samples from such inversion attacks. As suggested in the conclusion of (Geiping et al. 2020), the instance-level provable differential privacy may remain the only way to guarantee security for the individual training data sample in FL.
> >
> > [Zhu et al. 2019] Deep Leakage from Gradients. NeurIPS 2019
> >
> > [Geiping et al. 2020] Inverting Gradients - How easy is it to break privacy in federated learning? NeurIPS 2020
> >
> > [Shokri. et al. 2015]  Privacy-Preserving Deep Learning. CCS 2015.
> >
> > [Hitaj et al. 2017] Deep Models Under the GAN: Information Leakage from Collaborative Deep Learning. CCS 2017.

---

> > > ### Comment · Reviewer_oz1G · 2021-11-21
> > > **Response to rebuttal**
> > >
> > > I would like to thank the authors for the clarification. After reading the rebuttal, I decide to keep the original score. I currently do not have further questions or comments.

---

> > > > ### Author Response · Authors · 2021-11-24
> > > > **Further Response to Reviewer oz1G**
> > > >
> > > >
> > > > Dear Reviewer,
> > > >
> > > > Thank you for the very timely response. We would like to highlight that your comments have helped us to increase the readability and quality of the manuscript, and we have further improved our paper based on other reviewers’ comments. We note that you kept your score unchanged, thus we are wondering if we have addressed your concerns and if there are any other potential clarifications, improvements, or evaluations that you think would push this manuscript over towards the acceptance region? Is there something we were not able to address that we could clarify further?
> > > >
> > > > As you suggested in the original review, we carefully showed our difference with (Lecuyer et al. 2019)  and added such discussion in Appendix A.5. We explained the reason why we consider the “inherent DP -- robustness” connections in the FL setting (i.e., privacy protection is naturally required for FL) in the Introduction. Also, we evaluated the certifications of user/instance-level DPFL algorithms and compared the robustness of different existing user-level DPFL algorithms to provide more insights. Therefore, it would be very helpful to know how to progress/improve this manuscript further.
> > > >
> > > > Finally, we would like to bring more papers (Shokri. Et al. 2015, Hitaj et al. 2017, Malekzadeh, et al. 2021, Zhu et al. 2021) to your attention that focus on the instance-level DP under collaborative learning/federated learning. Indeed, the instance-level DP is an ongoing and open topic in FL, and we do hope the reviewer could consider our instance-level DP-robustness section as an independent contribution of our paper.
> > > >
> > > > Kind regards,
> > > >
> > > > Authors of Paper 599
> > > >
> > > >
> > > > **Reference**:
> > > > * [Shokri. et al. 2015] Privacy-Preserving Deep Learning. CCS 2015.
> > > > * [Hitaj et al. 2017] Deep Models Under the GAN: Information Leakage from Collaborative Deep Learning. CCS 2017.
> > > > * [Malekzadeh, et al. 2021] Dopamine: Differentially private federated learning on medical data. arXiv preprint arXiv:2101.11693, 2021.
> > > > * [Zhu et al. 2021] Voting-based approaches for differentially private federated learning, 2021.

---

### Official Review · Reviewer_Rayz · 2021-11-01

**Correctness:** 3
**Technical Novelty And Significance:** 2
**Empirical Novelty And Significance:** Not applicable
**Recommendation:** 3
**Confidence:** 4

**Main Review:**

Strong points:
i) The problem of data poisoning is an important one, and the connection between robustness and privacy is a nice one to use for getting provable guarantees.
ii) After doing some sporadic checking, I have not found any actual mistakes in the proofs.

Weak points (and comments/questions for the authors):
1) In my opinion the paper lacks focus: there are plenty of DPFL algorithms introduced, but very little testing and comparisons to existing work, while what I take as the main contribution, i.e., the certified robustness, is overshadowed by the FL parts and seems unfinished.

2) It is generally somewhat hard to tell which parts are meant as original contributions and which are referencing existing work (e.g. Sec 4.1, is this meant as original contribution or just paraphrasing existing work?)

3) The threat models of the proposed DPFL algorithms are not quite clear to me: e.g. for insDP, are the DP guarantees supposed to hold against adversaries who can poison some samples during the training? If so, this should probably affect the privacy guarantees resulting from subsampling in InsDP-FedSGD (since the adversary could have knowledge it the data partition in question has been chosen in the update?). How are the other DPFL adversaries?

4) The paper tends to oversell it's contribution:
4.1) The certified prediction Thm1 and it's proof match almost exactly with Prop1 from Lecuyer et al. 2019, the same goes for Thm3 and Cor1 compared to Ma et al. 2019 Thm4, and Cor6. Although both works are cited in the current paper, I do feel that this near-identity should be clearly and unambiguously stated in introducing these results. There is some discussion on this right before Section 6 noting similarities with Lecuyer et al., but stating that "ours focus on the robustness against training-time attacks in FL, which is more challenging considering the distributed nature and the model training dynamics, i.e., the analysis of the privacy budget over training rounds". But the federated setting only shows up in ascertaining that a training algorithm satisfies DP and in considering a suitable neighbourhood definition, it clearly does not complicate matters in the certified robustness Thms. If anything I would think that the no-show of any notion of federation in the proofs shows that at bottom these problems boil down to the ones considered by Lecuyer et al. & Ma et al. and are therefore not any harder. In general I find it a bit misleading that the results are introduced as somehow specifically concerning FL, when the certifiability results actually just use properties of DP, no matter if the setting is federated, otherwise distributed, or centralised; the main things are the neighbourhood definition (to determine what the adversary can control) and the privacy parameters.

4.2) As for the privacy accounting, since the proposed DPFL algorithms seem to be simple modification of existing ones, the privacy cost can be readily and accurately calculated using existing tools, and this seems to be exactly what the authors do. It is therefore hard to see what value does the hard-to-read Moments accountant type Prop.1 bring (note also that there exists a newer and much clarified paper on RDP, which results in improved bounds [1]).

References:
[1] Mironov et al. 2019: Rényi Differential Privacy of the Sampled Gaussian Mechanism

**Summary Of The Paper:**

## Update after rebuttal and discussions

I thank the authors for taking the time to discuss the issues pointed out in the reviews at length. Unfortunately, I am still not convinced that the paper is ready for publication. My main concerns:

1) There are now experiments in the updated paper claimed to be DP which are not (median clipping).

2) I continue to have doubts about the subsampling amplification. Simply stating that the sampling is random is not good enough, since the key issue is the added uncertainty due to the subsampling: if the sampling does not increase the adversary's uncertainty, there is no amplification. As an immediate remedy, I suggest the authors state the threat model more clearly.

3) I still think the paper can be improved a lot by taking the time to rewrite it focusing on the main contribution of certified robustness under DP and clarity of the presentation.

---

The paper looks at the robustness properties of differentially private (DP) federated learning (FL), focusing on learning classification models from labeled data. The main idea is to turn DP privacy guarantees into certifiable robustness properties. The authors look at 2 certifiable properties, namely, certified prediction (data poisoning does not alter most likely label), and certified attack cost (there is a lower bound on the loss the given attack tries to minimize). They continue to show that DP models in general guarantee these on some level that depends on the privacy bounds. The paper also presents several DPFL learning algorithms for user and instance-level DP.

**Summary Of The Review:**

While the paper seems to propose many different things, I find that these are mostly small variations on existing work and I think the authors tend to overstate the papers' contribution (see main review for more detailed comments). The paper generally also feels disorganized due to having several disparate topics (algorithms for training DPFL models, some results that would fit privacy accounting paper better, certifiable properties of DP models). In general, I think this paper could clearly benefit from more work to sharpen the focus and the contribution it has to make, and to make it more clear and readable as well.

---

> ### Author Response · Authors · 2021-11-21
> **Official Response to Reviewer Rayz (1/4)**
>
> We thank the reviewer for the valuable comments and detailed suggestions for improving the paper. We address the comments and add additional evaluations as follows.
>
> > Q1: There are plenty of DPFL algorithms introduced, but very little testing and comparisons to existing work.
>
> We thank the reviewer for the constructive comments to improve our paper.  The discussion of existing DPFL algorithms is presented in Section 2 “Related Work”.
>
> Following the reviewer's suggestion, in the revised version we included two more user-level DPFL works (Geyer et al., 2017; McMahan et al., 2018)  to study certified prediction with a total of four different DPFL methods given the same privacy budget $\epsilon$. Since all our proposed robustness certifications are agnostic to DPFL algorithms, i.e., certifications hold no matter how $(\epsilon, \delta)$ is achieved, we can compare the certified results based on different DPFL algorithms.  Specifically, we consider the following four DPFL algorithms:
> * flat clipping (UserDP-FedAvg) clips the concatenation of all the layers of model update with the L2 norm threshold $S$.
> * per-layer clipping (McMahan et al., 2018) clips each layer of model update with the L2 norm threshold $S$.
> * flat median clipping (Geyer et al., 2017) use the median of norms of clients’ model updates as threshold $S$ for flat clipping.
> * per-layer median clipping (Geyer et al., 2017)  use the median of each layer’s norms of clients’ model updates as threshold $S$  for per-layer clipping.
>
>
> For MNIST (CIFAR-10), we set $\epsilon$ to be 0.6319 (0.5346) which is obtained by training all DPFL algorithms with the same noise level $\sigma= 2.3$ ($\sigma=  3.0$) for the same number of rounds. For flat clipping and per-layer clipping, we set $S=0.7$ ($S=1$) on MNIST (CIFAR-10). Except for local epoch $E= 1$, other FL parameters setups are the same as in Table 1.
>
>
> Fig. 14 and Fig.15 in Appendix A.3.9 present the robustness certification results. On MNIST, the flat clipping is able to certify the largest number of adversaries $k$; while on CIFAR-10, the median clipping certifies the largest $k$ instead. Moreover, on both MNIST and CIFAR-10, flat clipping and per-layer clipping with the same $S$ lead to different certification results, while the results of flat median clipping and per-layer median clipping are nearly identical.
> This is because even under the same privacy protection $\epsilon$, different DPFL algorithms $\mathcal{M}$ produce trained models $\mathcal{M}(D)$ with different model performance, thus leading to different certified robustness. Specifically, in Theorem 2, given the same $\epsilon$ and $x$, $F_{\mathbb{A}}(\mathcal{M}(D),x)$ and $F_{\mathbb{B}}(\mathcal{M}(D),x)$ vary for different DPFL trained models $\mathcal{M}(D)$, thus causing different certified $\mathsf{K}$.
>
> The above results indicate that our proposed robustness certifications can serve as new metrics to benchmark the robustness of different DPFL algorithms, which can potentially motivate the investigation for better DPFL algorithms (i.e., different types of noise, clipping methods, subsampling strategies, or even different FL training protocols). We believe these analyses can provide new and important insights to the FL community.
>
> > Q2: The certified robustness, is overshadowed by the FL parts and seems unfinished. The paper generally also feels disorganized due to having several disparate topics.
>
> We are sorry to learn that the reviewer felt our paper was disorganized.
> Our paper is organized as 1) DPFL algorithm, 2) corresponding privacy analysis and 3) the inherent certified robustness properties, for both User DP and Ins DP sections. The introduction of DPFL is inevitable since we later need to quantify its fundamental connection to robustness via two robust criteria (i.e., certified prediction and certified cost).
>
> We followed your suggestion and improved our presentation for certified robustness:
> 1) We moved most of the user-level DPFL background from Section 4.1 to Appendix A.1 so as to emphasize the rest certified robustness part in Section 4.
> 2) We moved the instance-level DP definition from Section 5.1 to Appendix A.1
> 3) We added the certified robustness comparison results of different DPFL algorithms (the response to your Q1) to Appendix A.3.9 and mentioned it in Section 6.1 in the main paper.
> 4) We added the certified results on one language modeling task in Appendix A.3.8 and mentioned it in Section 6.1 in the main paper.

---

> > ### Author Response · Authors · 2021-11-21
> > **Official Response to Reviewer Rayz (2/4)**
> >
> > > Q3: Is Sec 4.1 meant as original contribution or just paraphrasing existing work?
> >
> > Sorry for the confusion and yes Sec 4.1 is meant to provide preliminaries on DPFL based on existing work. In our previous version of Sec 4.1, we first paraphrase the user-level DP, which is proposed in (McMahan et al., 2018) as we cite above Definition 2. Then the introduced UserDP-FedAvg (Algorithm 1) is one of the standard DPFL algorithms (Geyer et al., 2017; McMahan et al., 2018) as we cite. The detailed comparison to these two works is mentioned in the response to your Q1. Finally, the privacy guarantee in Prop. 1 is a modification of existing tools as we state “a generalization of (Abadi et al., 2016)”
> >
> > ​Section 4.1 is presented just as a necessary DPFL background for understanding the rest of Section 4 about DPFL-robustness, and we didn’t consider Section 4.1 user-level DPFL as our contribution.  Instead, as listed at the end of the introduction, our contributions mainly fall in
> > 1) two robustness criteria for quantifying the certified robustness in both levels of DPFL (Section 4.2, 5.2)
> > 2) improved Ins-level DPFL privacy guarantees (Section 5.1)
> > 3) empirical evaluation of the robustness criteria on two image datasets and one NLP dataset, and the robustness comparison of different User-level DPFL algorithms. (Section 6)
> >
> > To avoid possible confusion, we
> > 1) changed the title of Sec. 4.1 to be “User-level Privacy and Background.”
> > 2) moved the user-level DP definition and Proposition 1 to Appendix A.1
> > 3) simplified our introduction of UserDP-FedAvg, and compared it to the related work (Geyer et al., 2017; McMahan et al., 2018, Abadi et al., 2016) more clearly.
> >
> > > Q4: Hard to tell which parts are meant as original contributions and which are referencing existing work.
> >
> > Thanks for the helpful comment. Our original contributions are summarized at the end of the introduction. We have made it more clear by pointing out the corresponding section for each contribution in our revision.
> > ​
> >
> > > Q5: Threat models of the proposed DPFL algorithms are not quite clear. Will the fact that adversaries who can poison some samples during the training probably affect the privacy guarantees resulting from subsampling in InsDP-FedSGD? (since the adversary could have knowledge it the data partition in question has been chosen in the update?)
> >
> > We thank the reviewer for carefully checking the privacy guarantees and callout on the threat model.
> >
> > * For ins DP, in our threat model, we consider the attacker that *follows our training protocol* and has *no control* over which data partition (or batch) is sampled. Yes, the privacy guarantees are supposed to hold against adversaries who can poison some samples during the training. Note that we do not assume that the adversaries’ poisoning data always be sampled. In our algorithms, each batch is randomly subsampled, so the adversaries cannot control if poisoned data are sampled in each step.  Therefore in each step of InsDP-FedSGD, the sampling rate for each instance is $pq$, where $q$ is the user-level sampling probability and $p$ is the batch-level sampling probability. Also, different data partitions are sampled in each round, and the overall sampling rate is uniform across all rounds. Therefore, the privacy guarantee on the final model is correct regardless of the attacker’s observation on which data partition is sampled. The privacy guarantee analysis for InsDP-FedSGD can be found in the proof of Prop. 2 in Appendix  A.1.2.
> > * For user DP, in our threat model, similarly, we consider the attacker that follows our training protocol and has no control over which users are sampled.
> >
> > Sorry for the confusion, and we have added such discussion in Appendix A.2 “Threat Models”  to make it more clear.

---

> > > ### Author Response · Authors · 2021-11-21
> > > **Official Response to Reviewer Rayz (3/4)**
> > >
> > > > Q6: Comparison to Lecuyer et al. & Ma et al.
> > >
> > > We thank the reviewer for the insight on the comparison to Lecuyer et al. & Ma et al. We discussed the difference in both Section 2 “Related Work” and right before Section 6.
> > >
> > > In our work, the main motivation to analyze the certified robustness based on DP in FL framework is that FL naturally requires the privacy guarantees like DP and thus the robustness would be achieved “for free”. Compared to Lecuyer et al. & Ma et al., there are non-trivial differences in terms of the relationship between DPFL and robustness:
> > >
> > > * Lecuyer et al. work on certified prediction against *test-time* attacks while we study DP against *training-time* poisoning attacks in *FL*. We would like to emphasize that Lecuyer et al. aim to make the *classification process* Pixel-DP while the *training algorithm* itself does not satisfy DP, so it cannot directly build the relationship between DP and training-time certified robustness.
> > >   * Conceptually, in contrast to the connection between Pixel-DP and certified prediction against adversarial examples, our proposed analysis on the connection between DP and certified robustness against data poisoning is new.
> > >   * Technically, Pixel-DP adds noise on *test data samples* during testing *once* while we add noises in *updates/gradient* at *every training round*. Although the analysis of Pixel-DP and ours are both rooted in the intuition of DP definition,  Pixel-DP requires the classiﬁer to make randomized predictions during testing and *the training algorithm itself does not satisfy DP*. In contrast, our defense against data poisoning requires the **learning algorithm**, rather than the **classiﬁcation process** as in Pixel-DP, to be randomized. We add noise on the updates (user DP) or gradient (ins DP) over every training round to make the trained FL model satisfy DP, which requires careful privacy budget analysis of the DPFL model over multiple training rounds considering the “distributed nature and the model training dynamics”.
> > >   * Empirically, we explicitly evaluate the relationship between the privacy protection level $\epsilon$ and the certified robustness based on two robustness criteria on three datasets, indicating the fundamental connections quantitatively. Moreover, we compare the certified robustness of different DPFL algorithms (the response to Q1). We believe this is the first work to provide such insights in FL.
> > >
> > > * Ma et al. analyze the lower bound of certified cost in centralized DP learning while we use the certified cost as one of our certiﬁcation criteria for different definitions of DP in *FL* and additionally derive its *upper bounds*.
> > >
> > > The robustness certification against poisoning attacks in this paper requires rigorous and provable guarantees and it demands careful investigation when studying a new ML framework such as FL. We added the above discussion to clearly and unambiguously state the difference between our results and  Lecuyer et al. & Ma et al in Appendix A.5.

---

> > > > ### Author Response · Authors · 2021-11-21
> > > > **Official Response to Reviewer Rayz (4/4)**
> > > >
> > > > > Q7: But the federated setting only shows up in ascertaining that a training algorithm satisfies DP and in considering a suitable neighbourhood definition, it clearly does not complicate matters in the certified robustness Thms.
> > > >
> > > > We agree with the reviewer that federated setting is only analyzed when calculating the privacy guarantee $(\epsilon, \delta)$  for a specific DPFL algorithm, and the certified robustness Thms   generally hold for any FL models satisfying User/Ins $(\epsilon, \delta)$-DP.  This is because in our current Thm 1, 2, 3 and Corollary 1, we view the randomized DPFL mechanism $\mathcal{M}$ satisfying $(\epsilon, \delta)$-DP as a given function/system.
> > > >
> > > > However,  in practice, one can readily use our certifications to compare different DPFL algorithms satisfying the same $(\epsilon, \delta)$-DP, which could have different certified robustness results, as we show in the response to Q1. The optimal $\epsilon$ for certified prediction could vary under different DPFL algorithms as a tradeoff between utility and privacy.
> > > >
> > > >
> > > > > Q8:  Hard to see what value does the hard-to-read Moments accountant type Prop.1 bring.
> > > >
> > > > Thank the reviewer for the insightful comment.  We present the privacy guarantee Prop. 1 just for the completeness of our presentation, which follows the order of DPFL algorithm, corresponding privacy guarantee, and finally its robustness guarantee. We moved Prop.1 to Appendix A.1.1 in our revision to avoid potential confusion.
> > > >
> > > > Yes, Prop.1 can be improved based on RDP [1]. We mentioned [1] below Prop.1 in our revision.
> > > >
> > > >
> > > > > Q9:  Sharpen the focus and the contribution it has to make and to make it more clear and readable as well.
> > > >
> > > > We thank the reviewer for the valuable suggestion!
> > > > We summarize the contribution in the introduction and add the corresponding section number in our revision.
> > > >
> > > > Please let us know if you have other questions and comments, and we really look forward to discussing with the reviewer and further improving our paper.

---

### Official Review · Reviewer_u4ds · 2021-11-02

**Correctness:** 4
**Technical Novelty And Significance:** 2
**Empirical Novelty And Significance:** 3
**Recommendation:** 6
**Confidence:** 2

**Main Review:**

Update: Thank you for your response. I read the authors' response and other reviewers' comments. My second question has been well addressed. However, as other reviewers suggested, the contributions of this paper and comparisons with prior work should be explained more clearly. I would not consider the main contribution as the theory part, because given the group privacy property of DP, this result is not very surprising. I would suggest the authors highlight the empirical contributions and federated learning setting.


Strengths:
1. I think this paper provides an interesting perspective for robust machine learning.
2. The empirical evaluations are solid.
3. the proposed algorithms are practically useful


Weaknesses:
1. I think the results are correct but not surprising. There have been many papers (theoretically and empirically) showing either robust algorithms can be made private easily and private algorithms provide intrinsic robustness.

2. It would be better if the authors could provide some theoretical/empirical intuition for the utility. It is known that both robust learning algorithms and private algorithms would cause the performance drop. It would be nice if the authors could provide non-private(epsilon=infty) clean accuracy as a comparison.



**Summary Of The Paper:**

This paper studies differentially private federated learning and its intrinsic robustness against data poisoning attacks. Theoretically, the authors build two definitions for certified robustness against data poisoning attacks, draw the connection with user-level and instance-level differential privacy. The key proof is based on the definition of individual privacy and group privacy. Empirically, the authors verify the correctness of the bounds by performing real attacks. I think the main contribution is to establish the robustness bound.

**Summary Of The Review:**

This paper is overall well-written and provides new insight on intersection of robustness and privacy.

---

> ### Author Response · Authors · 2021-11-21
> **Official Response to Reviewer u4ds:**
>
> We thank the reviewer for the constructive comments and we provide our response below.
>
> > Q1: The results are not surprising. Contribution compared with prior DP-robustness works.
>
> Thanks for pointing this out.
>
> Although intuitively DP is related to robustness given the model smoothness realization, there is *no existing work* formally proving that DP leads to certified robustness against data poisoning attacks in *FL*. Here we take the ﬁrst step to *formally prove and quantify* such connections.
> * We first identify two levels of DP in FL, i.e., user-level and instance-level, and propose new mechanisms and analyses to achieve improved instance-level DP.
> * Then, to reveal DP-Robustness connections, we propose two quantitative criteria for certiﬁed robustness of FL against poisoning attacks, i.e., certified prediction and certified attack cost, *so that such connections can be built theoretically and evaluated practically*. These connections are not straightforward as FL has fundamentally different properties than traditional centralized ML frameworks (e.g., the notion of users, multiple attackers, two levels of DP definitions, and the corresponding different DPFL training algorithms).
> * Finally, we evaluate our proposed robustness certifications for DPFL algorithms on two image datasets and one language dataset.
>
>
> In addition, the comparison to prior works is discussed in Section 2 “Differential Privacy and Robustness” and Section 5.2 “Comparison with existing certiﬁed prediction methods in centralized setting”. Specifically,
> * Pixel-DP (Lecuyer et al.,2019) (a new DP definition on the pixel level) works on certified prediction against *test-time* attacks while we study DP against *training-time* poisoning attacks in *FL*. Note that Lecuyer et al. aim to make the *classification process* Pixel-DP while the *training algorithm* itself does not satisfy DP, so it cannot directly build the relationship between DP and training-time certified robustness, and it only focuses on the test-time robustness.
> * Ma et al., 2019 analyze the lower bound of certified cost via DP in *centralized* learning while we work on different definitions of DP in *FL* and additionally derive the *upper bound* of certified cost.
> * Other empirical works (Hong et al., 2020, Bagdasaryan et al., 2020; Sun et al., 2019) do not provide any theoretical robustness guarantee for DP.
>
> To further differentiate our contributions to existing works, we conducted a new experiment to apply our certification on different user-level DPFL models trained by different DPFL algorithms.  As shown in Figure 14 and Figure 15 of Appendix. A.3.9, the models trained by different user-level  DPFL algorithms satisfying the same privacy guarantee $\epsilon$ have different certified robustness results. It indicates that our proposed robustness certifications can serve as new metrics to benchmark the robustness of different DPFL algorithms, which can potentially motivate the investigation for better DPFL algorithms (i.e., different types of noise, clipping methods, subsampling strategies or even different FL training protocols). We believe these analyses can provide new and important insights to the FL community,
>
> > Q2: Provide some theoretical/empirical intuition for the utility. It is known that both robust learning algorithms and private algorithms would cause the performance drop.
>
> We thank the reviewer for the insightful comment. We first note that our certification does not cause additional performance reduction except for the DPFL training. Given the trained and fixed DPFL models satisfying User/Ins $(\epsilon, \delta)$-DP, we can derive the corresponding certifications, which do not require additional training steps, thus we call it “for free”.
>
> Following the suggestion, we report the clean accuracy (average over 1000 runs) of UserDP-FedAvg and InsDP-FedAvg under non-DP training ($\epsilon=\infty$) and DP training (the various $\epsilon$ used in our paper) on MNIST and CIFAR-10.  As shown in Table 2, 3, 4, 5 in Appendix A.3.2, compared to the non-DP training, the DP training does not cause significant performance reduction with the $\epsilon$ used in our paper.
>
> Theoretically, the utility of general DP DNN is an open and interesting area, and the utility analysis of the DPFL model is another challenging and interesting future work considering the distributed nature of FL.

---

> > ### Author Response · Authors · 2021-11-29
> > **Final Follow-up to Reviewer u4ds**
> >
> > Dear Reviewer,
> >
> > Thank you for your helpful feedback, and it has greatly helped us strengthen the paper. As the end of the discussion is approaching, we would like to summarize our updates following the suggestions below and we are happy to further improve the paper if there are additional comments.
> >
> > As suggested by the reviewer, we provide the comparison with prior DP-robustness work in our response and report the clean accuracy of DP-models and non-DP models in Appendix A.3.2. We have further improved our paper based on other reviewers’ comments, including the certified robustness on the Sentiment140 dataset (Appendix A.3.8) and a comparison of certified robustness for different existing DPFL methods (Appendix A.3.9). Other major updates of our revision are listed in our General Response.
> >
> > Thank you for your time and please let us know if you have any final questions that we can address.
> >
> > Kind regards,
> >
> > Authors of Paper 599

---

### Official Review · Reviewer_T4mM · 2021-11-03

**Correctness:** 3
**Technical Novelty And Significance:** 2
**Empirical Novelty And Significance:** 2
**Recommendation:** 5
**Confidence:** 4

**Main Review:**


Strength: the paper proposes criteria for robustness of FL, evaluates the theoretical results using MNIST and CIFAR datasets.


Weakness: I can't entirely agree with the paper's message that the certification comes for free and instead suggest that this certification comes at a relatively high price requiring differential privacy. For example, [1] uses gradient shaping without full DP. However, I would agree that in cases where differential privacy is inevitable, the conclusions seem helpful and could further promote the use of DP.

Furthermore, the paper does not evaluate the effects of certification on performance reduction for the benign data. Specifically, I am surprised by an extremely low budget in CIFAR for the presented experiments, i.e. epsilon less than 1 and an extreme amount of noise, i.e. std around 10.  Clarifying further A.3.2 that has only few rounds would be helpful.The provided certification might not provide much utility, especially for diverse users already disproportionally affected by less strict DP budget.

Evaluating the proposed method on the Language Modeling task might further strengthen the submission to provide a realistic privacy training regime.



[1] Sanghyun Hong, Varun Chandrasekaran, Yig ̆itcan Kaya, Tudor Dumitras ̧, and Nicolas Papernot. On the effectiveness of mitigating data poisoning attacks with gradient shaping. arXiv preprint arXiv:2002.11497, 2020.

**Summary Of The Paper:**

The paper states that the model produced by differentially private federated learning to be already certified against poisoning attacks.


**Summary Of The Review:**

The paper proposes a good analysis of provided robustness guarantees, but further clarification of experiments would be very helpful.

---

> ### Author Response · Authors · 2021-11-21
> **Official Response to Reviewer T4mM:**
>
> We thank the reviewer for the insightful comments. Below we provide our point-to-point response.
>
> > Q1: Certification comes at a relatively high price requiring differential privacy. For example, [1] uses gradient shaping without full DP.
>
> We thank the reviewer for pointing this out and actually we discussed [1] in Section 2 Related Work “Differential Privacy and Robustness”. We agree with the reviewer that only in cases where differential privacy is inevitable such as federated learning, such certified robustness would be “free” and this is also exactly what we aim to achieve. We mentioned this motivation in Section 1 Introduction.
>
> In particular, gradient shaping [1] empirically found that DP-SGD could increase the model’s robustness against data poisoning attacks, even in conﬁgurations that do not result in meaningful privacy guarantees ($\epsilon$ is large). Gradient shaping without full DP provides empirical robustness but does not offer any certified robustness guarantees.
> Instead, with DP, we can build robustness guarantees theoretically via two quantitative criteria for FL (i.e., certified prediction, and certified attack cost), and we further evaluate those criteria practically. Our theoretical guarantees derived from DP always hold under $k$ poisoning users/instances.
>
> Overall, in practice, FL systems indeed need privacy guarantees such as DP, so we focus on the connection between DPFL and its by-product: certified robustness.
>
> > Q2: The paper does not evaluate the effects of certification on performance reduction for the benign data.
>
> We thank the reviewer for the insightful comment.
>
> First, we would like to note that our certification does not cause additional performance reduction except for the DPFL training. Given the trained and fixed DPFL models satisfying User/Ins $(\epsilon, \delta)$-DP, we can derive the corresponding certifications, which do not require additional training steps, thus we call it “for free”.
>
> Following the suggestion, we report the clean accuracy (average over 1000 runs) of UserDP-FedAvg and InsDP-FedAvg under non-DP training ($\epsilon=\infty$) and DP training (the various $\epsilon$ used in our paper) on MNIST and CIFAR-10.  As shown in Table 2, 3, 4, 5 in Appendix A.3.2, compared to the non-DP training, the DP training does not cause significant performance reduction with the $\epsilon$ used in our paper.
>
>
>
>
> > Q3: Low budget in CIFAR for the presented experiments. Clarifying further A.3.2 that has only few rounds would be helpful.
>
> We thank the reviewer for pointing this out.
>
> As we mentioned in Appendix A.3.1,  “When training on CIFAR10, we follow the standard practice for differential privacy (Abadi et al., 2016; Jagielski et al., 2020) and ﬁne-tune a whole model pre-trained non-privately on the more complex CIFAR100”. We can achieve reasonable performance on CIFAR-10 datasets by training (ﬁne-tuning) few rounds, so it results in small $\epsilon$. We emphasized this in our revision in Appendix A.3.1 following your suggestion.
>
> Moreover, we note that our robustness certification is built upon the FL training regime that already satisfies differential privacy (DPFL). Due to the exponential property of DP definition, the $\epsilon$ cannot be large in general (Abadi et al.) for meaningful privacy guarantees.  Therefore, we use small $\epsilon$ to evaluate the robustness certifications.
>
> [Abadi et al.] Deep Learning with Differential Privacy. CCS 2016.
>
>  > Q4: Evaluating the proposed method on the Language Modeling task.
>
> We thank the reviewer for the great suggestion to improve our paper.
>
> We evaluate our robustness certification for a text sentiment analysis task on tweets from Sentiment140 (Sent140), which involves classifying Twitter posts as positive or negative, and is a language modeling task evaluated in standard FL paper (Tian et al. 2018).
> * FL setup: we use a two layer LSTM binary classiﬁer containing 256 hidden units with pretrained 300D GloVe embedding. Each twitter account corresponds to a device. We used the same network architectures, non-iid dataset partition method, learning rate, batch size, etc. as in (Tian et al. 2018), and those hyper-parameters are summarized in Table 1.
> * User-level DP setup: we set $\epsilon$ to be $0.2238, 0.2247, 0.4102. 0.7382, 1.7151$, which are obtained by training UserDP-FedAvg models with noise level $\sigma=4, 3, 2, 1.5, 1$, respectively.
>
> Figure 13 in Appendix. A.3.8. presents the user-level certified accuracy under different $\epsilon$ by training DPFL models with different noise scale $\sigma$. We observe that the largest $k$ can be certified when $\epsilon$ is around $0.2247$, which also verifies the tradeoff between $\epsilon$ and certified accuracy as we observed in image datasets.
>
> [Tian et al. 2018] Federated Optimization in Heterogeneous Networks. MLsys 2020

---

> > ### Comment · Reviewer_T4mM · 2021-11-23
> > **Thank you for the thorough response**
> >
> > > As we mentioned in Appendix A.3.1, “When training on CIFAR10, we follow the standard practice for differential privacy (Abadi et al., 2016; Jagielski et al., 2020) and ﬁne-tune a whole model pre-trained non-privately on the more complex CIFAR100”. We can achieve reasonable performance on CIFAR-10 datasets by training (ﬁne-tuning) few rounds, so it results in small
> > ϵ. We emphasized this in our revision in Appendix A.3.1 following your suggestion.
> >
> > I guess, I am a bit unclear on the threat model then. Why would the attacker want to compromise the model that is getting fine-tuned for just a couple of rounds. Given the scale and diversity of users this setting looks unrealistic to me.
> >
> > > As shown in Table 2, 3, 4, 5 in Appendix A.3.2, compared to the non-DP training, the DP training does not cause significant performance reduction with the ϵ used in our paper.
> >
> > Table 3 shows accuracy of 50-70% on the main CIFAR task which is hardly acceptable performance.
> >
> > > We evaluate our robustness certification for a text sentiment analysis task on tweets from Sentiment140 (Sent140), which involves classifying Twitter posts as positive or negative, and is a language modeling task evaluated in standard FL paper (Tian et al. 2018).
> >
> > I guess we are confusing terminology here: I refer to language modeling as a task of predicting the next or masked word (see [link](http://nlpprogress.com/english/language_modeling.html)). What you have done is just a classification task. The standard paper (please see [1] or [2]) uses exactly a language modeling task. Therefore, I wonder what guarantees you can obtain on a language modeling task given it's wide popularity in FL papers.
> >
> > Summary:
> >
> > Overall, I admit a good connection between Differential Privacy and Robustness, however I wonder whether the overall message of the paper that robustness comes for free is a valid claim. To me it looks like that in order to obtain robustness you need to impose extremely tight privacy budget (compared to [2] see Table 2), thus there are experiments that perform only fine-tuning of existing models to achieve eps<<1 budget, although they still harshly impact model performance. It looks like the claim could be rephrased as "Differential Privacy and Certified Robustness can be achieved using the same mechanism of clipping gradients and adding Gaussian noise".
> >
> > I appreciate authors contribution, however I will keep my score.
> >
> > References:
> >
> > [1] McMahan, B., Moore, E., Ramage, D., Hampson, S. and y Arcas, B.A.,. Communication-efficient learning of deep networks from decentralized data. In AISTATS'17
> >
> > [2] McMahan, H.B., Ramage, D., Talwar, K. and Zhang, L.,. Learning differentially private recurrent language models. ICLR'18.

---

> > > ### Author Response · Authors · 2021-11-24
> > > **Response to Reviewer T4mM (1/2)**
> > >
> > > We thank the reviewer for the further comments, and we provide our responses below.
> > >
> > > > Q1:  I guess, I am a bit unclear on the threat model then. Why would the attacker want to compromise the model that is getting fine-tuned for just a couple of rounds. Given the scale and diversity of users this setting looks unrealistic to me.
> > >
> > > Thank the reviewer for the comment. There could be some misunderstanding that we only consider fine-tuning as our threat model, which is not true.
> > > 1. In terms of our threat model, the attacker aims to poison the FL system that uses DPFL training, no matter if the training is fine-tuned or trained from scratch. The attacker's capability is the same for either case.
> > > 2. Our certification is agnostic to how the DPFL system is being trained - we can certify DPFL systems that are either trained from scratch (like MNIST and Sentiment140 setting) or fine-tuned (like CIFAR10 setting). The different training method (training from scratch or fine-tuning) is just a way to improve utility given the privacy budget. Fine-tuning on CIFAR to obtain DP is the standard way [1][2] including the most popular DP-SGD [1], and [2] exactly performs the data poisoning attack against a DP model that is finetuned on CIFAR. Training DP fully on a deep NN from scratch on complex tasks is still an open and ongoing topic. Note that although user-level DPFL in the industrial scale [3] can train private LSTM language models from scratch, it requires a total of $10^5$ and even $10^8$ (Table 2 of [3])  participants so as to maintain a low sampling ratio of every user, which is beyond our current computation capability.
> > >
> > > [1] Deep Learning with Differential Privacy. CCS 2016
> > >
> > > [2] Auditing differentially private machine learning: How private is private sgd? NeuIPS 2020.
> > >
> > > [3] McMahan, H.B., Ramage, D., Talwar, K. and Zhang, L.,. Learning differentially private recurrent language models. ICLR'18.
> > >
> > > > Q2: Table 3 shows accuracy of 50-70% on the main CIFAR task which is hardly acceptable performance.
> > >
> > > We thank the reviewer for pointing this out, but we believe there is an oversight of our submission.
> > > In table 3, the clean accuracy is 50-70% on CIFAR-10 only when $\epsilon$ is around 0.1, which corresponds to the first three lines in Figure 1 (b). For other lines in Figure 1 (b), the clean accuracy of corresponding $\epsilon$  is 70~80%.  We evaluate such extreme cases ($\epsilon$  is around 0.1) only to study the relationship between  $\epsilon$ and certified accuracy and to show the tradeoff between privacy and certified prediction.
> > >
> > > For other evaluations on our papers (Figure 2, 3, 4, 5), we use $\epsilon$  between 0.5~2, and the clean accuracy is ~ 80%. As shown in table 2, 81.90% is already the top-performance on this network trained without noise ($\epsilon=\infty$).
> > >
> > > > Q3:  I guess we are confusing terminology here: I refer to language modeling as a task of predicting the next or masked word (see link). What you have done is just a classification task. The standard paper (please see [1] or [2]) uses exactly a language modeling task. Therefore, I wonder what guarantees you can obtain on a language modeling task given it's wide popularity in FL papers.
> > >
> > > We apologize for the confusion about the language modeling task.
> > >
> > > * We note that [1] does not work on DPFL for their Shakespeare dataset.
> > > * [2] work on the user-level DPFL on the Reddit dataset. However, we note that in order to retain reasonable performance under reasonable $\epsilon$ (around 1 ~ 5 ), the authors of [2] have to train the LSTM model with a total of $10^5$  - $10^9$ participants (see Table 1 and Table 2 in [2]), thus maintaining a low user sampling ratio at each round. We are sorry that such industrial-scale experiments are beyond our computation capability.
> > > * [3] also points out the computational constraints so they can’t reproduce the results of [1].  They use 80000 total participants for Reddit dataset instead. They train 2000 or 3000 rounds to achieve similar accuracy (~18%), but they didn’t report the corresponding $\epsilon$ in their paper. We calculate their $\epsilon$ using moment accountant, which will be 1099853076.697097 after 2000 rounds when the user sampling ratio is 100/80000, the noise level $\sigma$ is 0.001 and $\alpha$ is 1e-6 (Please correct us if the calculation is wrong). Such large $\epsilon$ is meaningless in practice and therefore we don’t use such DPFL models to evaluate the certification. We will further clarify this point in the revised version.
> > >
> > > [1] McMahan, B., Moore, E., Ramage, D., Hampson, S. and y Arcas, B.A.,. Communication-efficient learning of deep networks from decentralized data. In AISTATS'17
> > >
> > > [2] McMahan, H.B., Ramage, D., Talwar, K. and Zhang, L.,. Learning differentially private recurrent language models. ICLR'18.
> > >
> > > [3] Bagdasaryan et al. Differential Privacy Has Disparate Impact on Model Accuracy. NeurIPS’19

---

> > > > ### Author Response · Authors · 2021-11-24
> > > > **Response to Reviewer T4mM (2/2)**
> > > >
> > > > > Q4: To me it looks like that in order to obtain robustness you need to impose extremely tight privacy budget (compared to [2] see Table 2), thus there are experiments that perform only fine-tuning of existing models to achieve eps<<1 budget, although they still harshly impact model performance.
> > > >
> > > > Thanks for the comment. We think there is a misunderstanding of our submission.
> > > > * First, we don’t need to impose an extremely tight privacy budget to obtain robustness. Actually, for certified prediction, there is a tradeoff between privacy and robustness so the extremely small eps is not the best.
> > > > * Second,  most of the experiments in our paper use $\epsilon$ = 0.5 - 2 budget, and we evaluate those extreme cases of “eps =~ 0.1” only to study the relationship between $\epsilon$ and certified accuracy in Figure 1.  Therefore robustness is not obtained just for “ an extremely tight privacy budget” nor “eps<<1”. Moreover, in the Table 2 of [1], there are 763430 users or $10^8$ users so as to obtain $\epsilon$ of 4.634~0.987, and we are very sorry that such industrial-scale evaluation is out of our computation capability for the rebuttal period. Also, our FL setting could be better fitted to FL on institutes like banks or hospitals, and the number of clients would be comparable to our current experiment setups.
> > > > * Finally, we would like to clarify that we are not sacrificing model performance for small eps to make the certification work better; we merely take any existing DPFL system to derive free robustness certificates. The bottleneck is the utility-privacy tradeoff of the DPFL algorithm itself; not our certification method.
> > > >
> > > >
> > > > [1] McMahan, H.B., Ramage, D., Talwar, K. and Zhang, L.,. Learning differentially private recurrent language models. ICLR'18.
> > > >
> > > > > Q5: I wonder whether the overall message of the paper that robustness comes for free is a valid claim. It looks like the claim could be rephrased as "Differential Privacy and Certified Robustness can be achieved using the same mechanism of clipping gradients and adding Gaussian noise”
> > > >
> > > > Thanks for the comment. The certified robustness theorems are built upon the “already trained (given)” DPFL models that satisfy ($\epsilon, \delta$)-DP, which is orthogonal to the type of noise (Gaussian noise or Laplacian noise) and the type of clipping (clipping gradients or clipping updates). Therefore, rephrasing as “the same mechanism of clipping gradients and adding Gaussian noise” is not appropriate, because our robustness certification result does not affect the DP training beforehand at all. One can readily use our certifications on any trained DPFL models, so we call it “for free”, and we will update our title as something like “uncovering the relationship between DP and certified robustness of FL against poisoning attack” to make it more clear.
> > > >
> > > > Please let us know if you have other questions and comments, and we really look forward to discussing with the reviewer and further improving our paper. Thank you!!!

---

> > > ### Author Response · Authors · 2021-11-29
> > > **Final Follow-up to Reviewer T4mM**
> > >
> > > Dear Reviewer,
> > >
> > > Thank you for your insightful comments and valuable time. Your comments have helped us to significantly strengthen the manuscript, and we have further improved our paper based on other reviewers’ comments. As the end of the discussion is approaching, we would like to ask if we have addressed your concerns and if there are any other potential clarifications or improvements that you think would be helpful in making your final evaluation?
> > >
> > > Following your suggestions in the original review, we 1) reported the clean accuracy of DP models and non-DP models; 2) explained that our certifications don’t require additional cost given the trained DPFL models; 3) evaluated the certified robustness on a new dataset (Twitter semantic analysis).
> > >
> > > However, we note that the reviewer might have some misunderstandings about our submission: 1) the certifications *do not* require extremely small eps but the reasonable eps commonly used to protect privacy. 2)  We evaluate extreme cases of small eps only to study the *tradeoff* between eps and certified prediction. 3) Our certification is *agnostic* to how the DPFL system is being trained (trained from scratch or fine-tuned), which only influences the utility of the model. We do not aim to make the certification work better; we merely take any existing DPFL system to derive free robustness certificates and show such inherent DP-Robustness connections in different levels of FL privacy. 4) We were sorry that we were unable to reproduce the Reddit language modeling results which involve $10^5 - 10^9$ users in (McMahan, et al. 2018)  given the computation constraints. Therefore, we presented the results on the Twitter semantic classification task and we really thank the reviewer for the suggestions to improve our paper.
> > >
> > > We answered each of your questions and updated the manuscript based on your feedback. We sincerely hope that the reviewer could consider our responses to your concerns. Thank you for your careful review and please let us know if you have any final questions that we can address!
> > >
> > >
> > > Kind regards,
> > >
> > > Authors of Paper 599

---

### Author Response · Authors · 2021-11-21
**General Response**

We thank the reviewers for the insightful comments and suggestions. We have made the following major updates in our revision accordingly, and we also answer each reviewer’s questions in detail.

* Improve the connection of the main contributions to the corresponding sections:
  * In the summary of our contributions at the end of the introduction, we added the corresponding section number for each contribution bullet.
* More results on robustness certification:
  * We added the certified accuracy result for UserDP-FedAvg on a language modeling task (Sentiment 140) and mentioned it in Section 6.1 in the main paper. The experimental setups for this new dataset are added in Section 6 and Appendix A.3.
  * We added the certified accuracy comparison results of four DPFL algorithms in  Appendix A.3.9 and mentioned them in Section 6.1.
  * We further clarified our threat models in Appendix A.2.
  * We added the comparison to DP-Robustness related work (Lecuyer et al., 2019a; Ma et al., 2019) in Appendix A.5.
* Clarify the DPFL parts:

  * We changed the title of Section 4.1 to be “User-level Privacy and Background.”
  * We moved the user-level DP definition and Proposition 1 from Section 4.1 to Appendix A.1.
  * We simplified our introduction of UserDP-FedAvg (Algo. 1) in Section 4.1 and compared it to the related work more clearly.
  * We moved the instance-level DP definition from Section 5.1 to Appendix A.1.

* Clean accuracy:
  * We added four tables in  Appendix A.3.2 to report the clean accuracy of different algorithms and datasets.


All of our revisions are updated in OpenReview now and highlighted in blue. Thank you!

---

### Decision · Program_Chairs · 2022-01-20

**Decision:**

Reject

**Comment:**

The premise is an exciting observation: Differential privacy in federated
learning might imply being certified against poisoning attacks. While
this may be considered not surprising by some, the connection between
differential privacy and robustness is interesting to many. The
relationship was characterized both theoretically and empirically.

The reviewers discussed the paper extensively with the authors, and
while many issues were clarified, issues on correctness still
remained: it is unclear if the proposed DP mechanism actually is DP,
and subsampling amplification also had issues. Clarity needs to be
added in the writing, and the extensive comments by the reviewers
hopefully help the authors in that.